# Evaluating Thromboprophylaxis Strategies for High-Risk Pregnancy: A Current Perspective

**DOI:** 10.3390/ph17060773

**Published:** 2024-06-13

**Authors:** Lucia Stančiaková, Kristína Brisudová, Ingrid Škorňová, Tomáš Bolek, Matej Samoš, Kamil Biringer, Ján Staško, Juraj Sokol

**Affiliations:** 1National Center of Hemostasis and Thrombosis, Department of Hematology and Transfusion Medicine, Martin University Hospital, Jessenius Faculty of Medicine in Martin, Comenius University in Bratislava, 036 01 Martin, Slovakia; inkaskornova@gmail.com (I.Š.); jan.stasko@uniba.sk (J.S.); juraj.sokol@uniba.sk (J.S.); 2Department of Internal Medicine I., Martin University Hospital, Jessenius Faculty of Medicine in Martin, Comenius University in Bratislava, 036 01 Martin, Slovakia; kgrilus@gmail.com (K.B.); ato.bolek@gmail.com (T.B.);; 3Department of Gynecology and Obstetrics, Martin University Hospital, Jessenius Faculty of Medicine in Martin, Comenius University in Bratislava, 036 01 Martin, Slovakia; kamil.biringer@uniba.sk

**Keywords:** risk factors, at-risk pregnant patients, venous thromboembolism, thromboprophylaxis

## Abstract

Venous thromboembolism (VTE) represents one of the leading causes of death during pregnancy. The greatest risk for it is the presence of medical or family history of VTE, stillbirth, cesarean section and selected thrombophilia. Appropriate thromboprophylaxis has the potential to decrease the risk of VTE in at-risk pregnant patients by 60–70%. Based on this, the authors reviewed the PubMed, Web of Science and Scopus databases to identify the possibilities of thromboprophylaxis in pregnant patients with a high risk of VTE. Moreover, they summarized its management in specific situations, such as cesarean delivery or neuraxial blockade. Currently, low-molecular-weight heparins (LMWH) are the preferred drugs for anticoagulant thromboprophylaxis in the course of pregnancy and postpartum due to easy administration and a lower rate of adverse events.

## 1. Introduction

The first well-described case of deep vein thrombosis (DVT) was documented in the Middle Ages, when in 1271, Raoul noticed a unilateral edema of the ankle extending to the leg. During the Renaissance, pregnancy-related DVT was one of the leading causes of DVT at that time. It was presumed that postpartum DVT was developed due to the retention of unconsumed milk in the lower extremities [1].

Currently, the risk of venous thromboembolism (VTE) is higher in women than in men. Along with the use of combined oral contraceptives, pregnancy belongs to the clinical states increasing the risk of thromboembolic complications in women of childbearing age [2].

Pregnancy influences all three elements of the Virchow’s triad: hypercoagulability, endothelial dysfunction and hemodynamic alterations (Figure 1) [3].

Hypercoagulability during pregnancy is a consequence of increased or decreased activity and quantity of coagulation factors, and it is increased 5–10-fold in comparison to nonpregnant levels [4]. There is an increased activity of fibrinogen, coagulation factors VII (FVII), VIII (FVIII), IX (FIX), X (FX), XII (FXII), XIII (FXIII) and von Willebrand factor (vWF). Approximately 50% more molecules of fibrinogen are produced. Levels of factor II (FII) and V (FV) do not change significantly during pregnancy, and factor XI (FXI) activity decreases by 20–30% of up to 40% of the reference range. As one of the most important natural inhibitors of coagulation, antithrombin is not affected by hormones. It only slightly decreases by 15% in the last weeks of pregnancy. The synthesis of protein C (PC) is different depending on the gestational age, with an increase in the second trimester, a decrease in the third trimester, and another increase during the immediate postpartum period. There is a gradual decrease in protein S (PS) function, continuing for 2 months postpartum and longer in the case of breastfeeding. Regarding the fibrinolytic system, D-dimers increase throughout the pregnancy. The increased formation of thrombin is maximal at term and contributes to the prevention of massive hemorrhage during the delivery. It is associated with a significant increase in the levels of markers of coagulation activation (fragments of prothrombin 1+2 and thrombin–antithrombin complexes). Pregnancy provokes a decrease in tissue plasminogen activator (tPA), an increase in plasminogen activator inhibitor (PAI)-1 activity and the synthesis of PAI-2 by the placenta. Conclusively, there is the development of hypofibrinolysis [5]. The levels of novel coagulation markers plasmin-α2 plasmin inhibitor complex (PIC), thrombin–antithrombin complex (TAT), thrombomodulin (TM) and tissue plasminogen activator/plasminogen activator inhibitor compound (tPAI-C) in pregnant patients increase significantly throughout the pregnancy and gradually return to normal ranges after delivery. Their reference intervals in healthy pregnant women by trimester were established according to Clinical and Laboratory Standards Institute Document C28-A3c, thus contributing to the judgement of prothrombotic risk [6]. Additionally, after the first trimester, platelet aggregation also increases [7]. Figure 2 provides a summary of the influence of pregnancy on the markers of hemostasis [8].

The endothelium is composed of the cells lining the blood vessels. Endothelial cells are covered by the endothelial glycocalyx (EG), which prevents platelet and leukocyte adhesion, provides antithrombotic activity and also maintains tissue integrity. It is located on the luminal surface of endothelial cells. Injury of the EG is related to clinical conditions like inflammation, thrombosis, preeclampsia or fetal growth restriction (FGR) [9].

Uteroplacental circulation is formed at the beginning of the second trimester. The functional adaptation of uterine arteries and the remodeling of spiral arteries cause a change in the uteroplacental circulation to a high-flow and low-resistance system. This process is supported mainly by estrogens [10].

Described changes of circulation in pregnancy lead to a variety of changes in pharmacokinetics of anticoagulants. This is due to an increase in maternal plasma volume that results in increased distribution volume for water-soluble drugs and reduced peak and steady-state drug concentrations. There is also an increase in renal blood flow, and thus an increased glomerular filtration rate, from the second trimester, leading to quicker clearance of drugs excreted by the kidneys. Last but not least, during pregnancy, we can observe an increase in the free fraction of highly protein-bound agents because of lower concentrations of albumin in pregnancy associated with the placental and fetal metabolic effects [7].

VTE represents a leading cause of death of pregnant and postpartum women. The risk of VTE is higher in the course of pregnancy when compared with the nonpregnant state and peaks in the postpartum period: pooled incidence rates are 1.2 (95% confidence interval (CI): 1.0–1.4) and 4.2 (95% CI: 2.4–7.6) per 1000 persons [11]. The pooled incidence rate for DVT is 1.1 (1.0–1.3) per 1000 patients and 0.3 (0.2–0.4) per 1000 persons for pulmonary embolism (PE) [12]. However, the VTE risk remains increased during the 12 weeks after delivery. The majority of the cases are attributed to DVT, but PE is of greater concern, because it causes 10–15% of all pregnancy-associated maternal deaths in developed countries [13].

The cornerstone of the prevention of VTE-related maternal death is a mechanic and pharmacologic thromboprophylaxis. The history of the management of VTE is outlined in Figure 3 [1,14].

The use of low-molecular-weight heparin (LMWH) has the potential to reduce the risk of VTE by 60–70% in at-risk pregnant patients [15].

Based on this knowledge, the authors aimed to summarize the current approaches for thromboprophylaxis during at-risk pregnancy and to fill the gap in the existing literature regarding the optimal dose of thromboprophylaxis, the duration of an increased risk of VTE and the interaction of risk factors of thromboembolic complications in pregnancy. We also examined specific aspects of thromboprophylaxis in high-risk pregnancy, such as possibilities of monitoring its effectiveness, or the use of thromboprophylaxis in the setting of cesarean section and neuraxial anesthesia.

## 2. Methods

This article is written in the form of a non-systematic (narrative) review. We performed a search in the medical literature databases PubMed Central, Web of Science and Scopus. To identify the most appropriate articles from the recent sources that were processed in the form of presented narrative review articles, we used the keywords “pregnancy”, “venous thromboembolism” and “anticoagulant thromboprophylaxis”.

Wherever possible, we aimed to review the information from articles published during the last five years. We included articles found by searching the databases that were written mainly in the English language or translated to English from another original language. We excluded all the studies performed only in a nonpregnant population. Regarding quality assessment, we preferred information obtained in randomized controlled trials and systematic reviews.

## 3. Which Women Are “At Risk” and Why?

### 3.1. Previous Thromboembolic Episode

The incidence of recurrent VTE during pregnancy is approximately 7.6% [16,17,18], and this risk is increased more than 3-fold when compared with a nonpregnant state. However, this information was obtained in a retrospective study performed only in 109 patients who had at least one pregnancy following VTE [19]. The risk of antepartum recurrent VTE is increased in patients with a history of two or more previous thromboembolic events, antiphospholipid syndrome or paradoxically long-term anticoagulation. It is disputed that antepartum thromboprophylaxis with even an intermediate dose of LMWH might not be sufficient in this population [17]. Recurrent VTE is a potentially life-threatening clinical condition that increases the probability of the development of a post-thrombotic syndrome [19].

### 3.2. Family History of Venous Thromboembolism

Several studies have shown that positive family history of VTE along with thrombophilia are strong risk factors for the development of VTE, and especially PE, in women during pregnancy. These factors increase the prothrombotic risk from 3.7-fold to 8.5-fold [20,21,22,23].

Some studies providing this information are retrospective [20], with features of post hoc analyses [22]. Potential confounding with other unmeasured factors could not be considered here [20], and the presence of a control group might be beneficial [22].

On the other hand, the report of Jerjes-Sánchez et al. is based on a multicentric study performed in patients with VTE associated with pregnancy included in the Global Anticoagulant Registry in the FIELD (GARFIELD)-VTE [21]. Additionally, the systematic review of Nanne Croles et al. identified the largest number of studies investigating thrombophilia and pregnancy-associated VTE, with the inclusion of 41,297 pregnancies. It did not take into account changes in the methodology of diagnosis of thrombophilia [23].

### 3.3. Smoking Prior to or during Pregnancy

Tobacco increases the risk of VTE by several mechanisms—it raises levels of fibrinogen, tissue factor, FII, FV, FVIII, FX, FXIII and homocysteine [24]. Smoking is also a risk factor of pregnancy-associated VTE [25]. The adjusted odds ratio (AOR) for smoking 10–30 cigarettes/day before or during the course of pregnancy as a risk factor of VTE is 2.1 (95% CI 1.3–3.4) [26].

### 3.4. Obesity

It is suggested that obesity predisposes to venous stasis and alters the coagulation system by impairment of the fibrinolytic activity and an increase in the levels of D-dimers, fibrinogen, FVIII and FIX. Additionally, overweight patients have higher concentrations of C-reactive protein, promoting low-grade systemic inflammation [27]. Women with a BMI of 30–35 kg/m^2^ (category of high normal BMI) have an adjusted hazard ratio (HR) of 2.35 (95% CI 2.04–2.70), and patients with ≥35 kg/m^2^ (considered as severely obese) have a HR of 3.47 (95% CI 2.82–4.25). Therefore, there is a significantly increased long-term risk of post-pregnancy VTE [28].

These results were obtained in a study that included a large study population of 1,068,040 patients and near-complete coverage with a long-term prospective follow-up. However, the authors did not consider subsequent pregnancies and changes in BMI during follow-up. A further limitation is the fact that the diagnoses were not verified [28].

### 3.5. Age

The risk of VTE increases with age. It is correlated with a simultaneous enhancement of coagulation activation, represented by increased levels of coagulation activation peptides and decreased fibrinolytic activity—states aggravated by the existence of prothrombotic comorbidities [29]. The risk of postpartum VTE is increased in women aged over 35 years. On the contrary, age ≥ 40 years (adjusted OR 1.67) is a risk factor of PE in the postpartum period to a lesser extent. Before undergoing IVF, it is recommended to assess the risk factors of thromboembolic complications, including increased age (>40 years) [30].

### 3.6. Parity with Three or More Pregnancies

Multiparity with a history of three or more pregnancies was evaluated as one of the most important risk factors for VTE not only during hospitalization in pregnancy (OR 3.5, 95% CI 3.0–4.0) [31,32], but also throughout the pregnancy [33].

However, the incidence of VTE might have been higher due to the impossibility of contacting some of the patients [31].

Multiple pregnancies increase the risk of VTE [31,34,35,36,37], with an aOR of 2.7 (95% CI 1.6–4.5) [32]. However, some studies providing this fact were retrospective [35,36] or performed in the form of survey [34].

### 3.7. Anemia

Iron deficiency is a relatively unknown but also important risk factor for VTE. It induces thrombocytosis, which predisposes to hypercoagulability [38]. The adjusted OR for VTE is 2.6 (95% CI is 2.2–2.9) [26].

### 3.8. Varicose Veins

Varicose veins are typical for intimal hyperplastic areas and plaque that is infiltrated with leukocytes and mast cells. In pregnancy, type III collagen and elastin are reduced, and elastin fibers are fragmented [39]. The resulting abnormal blood flow and endothelial damage leads to an increased expression of tissue factor, PAI-1 and vascular cell adhesion molecule 1 (VCAM-1). Simultaneously, we can detect the activation of platelets, leukocytes, tissue factor-positive microvesicles and neutrophil extracellular traps (NETs) [40]. In the pathogenesis of varicose veins, further molecules have been involved: FoxC2-Delta-like ligand 4 (Dll4) pathway signaling; lnc-RNA, affecting the proliferation and migration of human saphenous vein smooth muscle cells through annexin A2; hypoxia-inducible factor target genes, such as GLUT1, CA9, vascular endothelial growth factor (VEGF) and BNIP3, which are upregulated; increased matrix metalloproteinase 2, 9 and 13 expression; tissue inhibitors of these matrix metalloproteinases; increased transforming growth factor β levels and dysregulation in the expression of the molecules, such as endothelin 1; laminin, intercellular adhesion molecule 1 (ICAM-1); endothelial cell-leukocyte adhesion molecule 1 (ELAM-1); angiotensin-converting enzyme, L-selectin; and proapoptotic regulators Bax and poly-ADP-ribose polymerase or cyclic guanosine monophosphate [41].

Varicose veins are a frequent finding in women with two or more pregnancies. Enhanced engorgement and impaired blood flow through them during pregnancy predispose the patient to thrombophlebitis [42]. Gross varicose veins were identified as a significant and often underestimated risk factor of VTE, with aOR of 2.69 (95% CI 1.53–4.7) [20,32,34,43,44]. A strength of the study by Sultan et al. [44] is the large database of patients, containing 376 154 pregnancies between the years of 1995 and 2009. However, it lacked validation studies of obstetric complications, and there was a scarcity of data estimating their prevalence in the global population. Moreover, data from other mentioned studies were retrospective [42] or obtained in a survey format [34].

### 3.9. Preeclampsia

Preeclampsia occurs in up to 8% of pregnancies. The risk of VTE is attributed to the increased expression of procoagulant factors, attenuation of endogenous natural inhibitors of coagulation, endothelial dysfunction, and enhanced platelet activity, and it is most prominent in the postpartum period [45]. As further pathogenetic characteristics, the function of cell-free DNA and NETs was recently proposed [46]. This fact is useful because of the involvement of agents such as vitamin D, nuclear factor kappa B inhibitors and acetylsalicylic acid in the treatment of NETosis in preeclampsia [47]. Attention should also be paid to the fact that preeclampsia and arterial hypertension increase the risk of VTE in the course of pregnancy and the postpartum period and even in the 13 years after [48,49]. The aOR for the risk of VTE in these patients is 3.1 (95% CI 1.8–5.3) [32]. The limitations of the study providing this conclusion were the inclusion criteria of patients with VTE during pregnancy who received LMWH, which could have contributed to an underestimation of the incidence of thromboembolic complications; a lack of information about potential death or arterial thrombotic events; and an inconsistent definition of hypertension during pregnancy and preeclampsia [48].

Additionally, the second of these studies did not include pregnancy loss developed before 20 weeks of gestation. Moreover, data on ultrasonography, blood samples and history of the patients were not fully available [49].

Preeclampsia is also considered to be a risk factor for arterial diseases [50].

### 3.10. Weight Gain

A retrospective study by Wu et al. showed that higher weight gain in the course of pregnancy contributes to increased risk of VTE [51], especially when it is more than 21 kg, with aOR of 1.6 (95% CI 1.1–2.6) [26,32].

### 3.11. Stillbirth

Stillbirth is another feature associated with an increased risk of VTE [52]. It is one of the strongest predictors of this complication in the postpartum period (incidence rate ratio (IRR) 6.2) [44,52,53,54].

### 3.12. Preterm Delivery

The delivery of a newborn before term is considered to be an intermediate or even strong risk factor for maternal postpartum VTE [44,55]. The strengths of this study were its population-based design, the high validity of newborn birth weight measurements and the sample size; limitations were the inclusion of hospitalized events, a lack of data about thrombophilia and thromboprophylaxis, the possibility of residual confounding and incomplete data on BMI [55]. Preterm delivery increases the risk of thromboembolic complications by approximately three times (aOR 2.7, 95% CI (2–6.6)) [26].

### 3.13. Cesarean Section

Cesarean section increases the risk of VTE. There are also further factors, such as prothrombotic effects of surgical delivery lasting more than 30 min or weight gain during pregnancy [56]. When compared with women who had a spontaneous vaginal delivery, cesarean section increases the risk of VTE by four-fold [57], and the aOR is 2.1 (95% CI 1.8–2.4) [32].

### 3.14. Peripartum Hemorrhage

Postpartum hemorrhage is considered to be significant when the estimated volume of blood loss is more than 1000 mL, and this represents an aOR of 4.1 (95% CI 2.3–7.3) [26,58]. However, the data on immobilization have only been addressed in a limited number of case-control studies up to now [58].

### 3.15. Postpartum Infection

The incidence of postpartum infection increases with the presence of smoking, employment, obesity, BMI > 30, anemia, diabetes mellitus, age of the woman being more than 35 years, the degree of the perineal wound, the use of equipment for delivery and its duration and the frequency of vaginal examinations. It confers an aOR for VTE of 4.1 (2.9–5.7). There have been a limited number of studies investigating the effectiveness of antibiotic prophylaxis in women following vaginal delivery [32,59]. In the puerperium, infections increase the risk of thrombosis by four-fold [60].

### 3.16. Transfusion

Red blood cell transfusion represents a relative risk of VTE of 3.9, and the administration of other transfusions or infusions of procoagulant fractions represents a risk of 8.2 [61].

### 3.17. In Vitro Fertilization

The hyperestrogenism resulting from ovarian stimulation causes an increase in the activity of procoagulant factors, a decrease in the levels of natural anticoagulants and reduced fibrinolytic capacity. It is rare to develop VTE before the use of chorionic gonadotropin (hCG). Only 3% of arterial thrombosis and VTE occurred before the administration of final oocyte maturation trigger [62]. The procedure of in vitro fertilization (IVF) is associated with an aOR of 2.7 (95% CI 2.1–3.6) [32]. Therefore, several prothrombotic risk factors during the IVF cycle in the development of VTE have been identified—aggressive stimulation, increased estradiol levels, the use of hCG and exogenous estrogen for frozen embryo transfer, ovarian hyperstimulation syndrome (OHSS) and multiple pregnancies [62].

### 3.18. Ovarian Hyperstimulation Syndrome

OHSS is the most severe complication of IVF. Young age, polycystic ovarian syndrome (PCOS), low BMI, a rapid increase in serum estradiol, a higher number of growing follicles at the time of triggering and many oocytes retrieved are its risk factors. OHSS consists of several components, such as ovarian enlargement, fluid shift into the third space and its resulting hemoconcentration, hypovolemia, renal failure, serosal effusions and hypercoagulable state [63]. The risk of VTE in patients undergoing IVF complicated by OHSS is increased by 100 times when compared to background risk [64]. The aOR for VTE is 2.7 (2.1–3.6) [32].

### 3.19. Antepartum Immobilization

Immobility increases the risk of VTE by 1.5–2-fold [60]. Such risk, related to the venous stasis, is dependent on the BMI of pregnant woman—when her pre-pregnancy BMI is ≥25 kg/m^2^, the aOR is 62.3 (95% CI 11.5–337). When pre-pregnancy BMI is <25 kg/m^2^, the aOR is 7.7 (95% CI 3.2–19) [26,32].

Generally, there is still a lack of adequate study data. Strategies for the prevention of VTE in pregnancy are often deduced from case-control or observational studies, with extrapolations from suggestions proposed for non-pregnant patients. A complex, multidisciplinary approach is thus often required, predominantly in the peripartum period [26].

## 4. When to Use Antithrombotic Prophylaxis

The risk of thromboembolic complications should be evaluated during pregnancy and also in the postpartum period.

Different guidelines provide various stratification systems for the need for thromboprophylaxis, and a summary of these is outlined in Table 1 [65,66,67,68].

Furthermore, according to the Asian venous thromboembolism guidelines adapted from the guidelines of the American College of Chest Physicians (ACCP), the American Heart Association and the American Stroke Association (AHA/ASA), the American Society for Metabolic and Bariatric Surgery (ASMBS), the RCOG and the European Society for Medical Oncology (ESMO), patients with four or more risk factors ought to be evaluated for a prophylactic dose of LMWH throughout the pregnancy and for six weeks postpartum [69].

Thus, there is a difference in the situation when the patient has a history of unprovoked VTE or an episode of VTE associated with transient risk factors (Table 2) [70].

## 5. How Important Is the Presence of Thrombophilia and Why?

Table 3 provides a summary of the guidelines focused on the prevention of VTE in pregnant women with inherited thrombophilia [32,71].

In the case of the co-existence of thrombophilic states, ASH guidelines recommend thromboprophylaxis either during ante- and postpartum period regardless of family history of VTE [71].

## 6. What Is the Form of Anticoagulant Thromboprophylaxis and Which Dose Should Be Used?

Generally, apart from sufficient fluid intake and exercise, ways of non-pharmacologic prophylaxis of VTE are the use of thromboembolic-deterrent stockings, graduated compression stockings and an intermittent pneumatic compression or sequential compression device. However, there is limited evidence related to the use of these aids in the pregnancy and postpartum period. Before prescribing pharmacologic thromboprophylaxis, several features have to be taken into account (Table 4) [67].

### 6.1. Unfractionated Heparin

#### 6.1.1. Pharmacokinetics in Pregnancy

In pregnant patients receiving a single weight-based dose of UFH (9500 ± 640 IU), a lower peak plasma concentration (0.11 ± 0.017 vs. 0.23 ± 0.036 IU/mL) and a shorter time to such peak plasma concentration (113 ± 20 vs. 222 ± 20 min) were detected. Plasma heparin concentration designated as the area under the curve for pregnant patients is 55% that of the values of nonpregnant subjects. Moreover, pregnant women have a non-significant increase from their baseline activated partial thromboplastin time (aPTT). Additionally, no significant increase in anti-Xa activity in pregnant patients administered a single dose of UFH 7500 IU before cesarean delivery and after NB can be measured.

These findings might be attributed to increased binding of circulating heparin-neutralizing proteins and increased levels of FVIII and fibrinogen in pregnant patients. Moreover, in the third trimester, there might be a higher heparin requirement, or potentially a lower requirement, because of a decrease in placental heparinase activity due to placental aging. Conclusively, in pregnant patients treated with a single dose of UFH 5000–10,000 IU, the peak plasma heparin concentration might be at or below the plasma heparin concentration detected 6 h after this UFH dose in nonpregnant individuals [7].

#### 6.1.2. Dosages

An outline of the dosing regimens of UFH is provided in Table 5 [70] and Table 6 [67].

### 6.2. Low-Molecular-Weight Heparin

#### 6.2.1. Pharmacokinetics in Pregnancy

The group of drugs entitled LMWH possess favorable properties, such as easy administration, an improved bioavailability and safety and more predictable pharmacokinetics. Moreover, a decreased occurrence of osteoporosis, heparin-induced thrombocytopenia and bleeding complications in comparison to UFH were reported [7]. Further advantages of LMWH include a longer half-life, predictable treatment response and less allergic reactions [70].

#### 6.2.2. Enoxaparin

The volume of distribution and its clearance increase during pregnancy. The first of these characteristics increases 49%, with the largest proportion in the third trimester, and resolves a few days after delivery. Clearance increases 48% with changes in glomerular filtration rate and normalizes 2 weeks following delivery. It seems that the peak anti-Xa activity is lower in pregnant patients in comparison to nonpregnant women. The maximum plasma concentration and anti-Xa activity are lower in the course of pregnancy when compared to the levels achieved six weeks after delivery [7].

#### 6.2.3. Dalteparin

The mean maximal concentration (0.21 ± 0.05 vs. 0.49 ± 0.13 anti-Xa IU/mL) and area under the curve (1.97 ± 0.46 IU vs. 3.23 ± 0.85 h/mL) detected up to 24 h from the administration of this drug are also lower when compared to nonpregnant women. The average time to achieve peak concentration is 3 h. The mean half-life for a morning dose is 4.9, and for evening dose, it is 3.9 h. In pregnancy, there are significantly lower mean anti-Xa activities in comparison to postpartum, with the lowest level at the 36th week of gestation [7].

#### 6.2.4. Tinzaparin

Peak anti-Xa activity measured 4 h after its administration is commonly below 0.1 IU/mL, and 24 h anti-Xa activity detected at 36 weeks of gestation compared to 28 weeks is reduced [7].

A summary of dosages of particular LMWHs is provided in Table 7 [65,66,67,70,72,73,74].

#### 6.2.5. Protamine Sulfate to Reverse Anticoagulation with Low-Molecular-Weight Heparin and Unfractionated Heparin

Protamine sulfate might lead to full reversal of UFH and 60–80% reversal of LMWH. One milligram of intravenous protamine can neutralize 100 IU of intravenous heparin. Reversal of subcutaneous heparin might require repeated doses of protamine (its half-life is approximately 7 min). Adverse effects in pregnant women are hypotension from histamine release, hypersensitivity reaction or even anaphylaxis, noncardiogenic pulmonary edema, pulmonary hypertension, thrombocytopenia, impaired platelet aggregation, decreased thrombin effect and fibrinogen precipitation [7].

### 6.3. Fondaparinux

This pentasaccharide crosses the placental barrier, so it is not recommended during the first trimester. However, reports of its use in more advanced stages of pregnancy exist. It is mainly used in pregnant patients with contraindication to heparin due to thrombocytopenia or allergic reaction. Moreover, fondaparinux is considered safe during breastfeeding [70,76]. The suggested dose according to clinical trials performed in nonpregnant women is 2.5 mg daily [72].

### 6.4. Danaparoid

Danaparoid represents another heparanoid molecule that is—like fondaparinux—recommended in pregnant women with allergy to LMWHs or HIT, because it does not cross-react with HIT antibodies [72]. Additionally, this LMWH derivative does not cross the placenta [65]. According to the SOGC guidelines, its recommended prophylactic dose in pregnancy is 750 IU administered subcutaneously twice a day [72].

### 6.5. Vitamin K antagonists

Vitamin K antagonists (VKAs) are able to cross the placental barrier and lead to fetal abnormalities (member and nasal hypoplasia, scoliosis, chondral calcification, schizencephaly and fetus intracranial hemorrhage) from the 6th week of gestation. Additionally, their use in the third trimester contributes to peripartum fetal hemorrhage, and its administration in the second trimester is correlated with neurologic impairment. However, warfarin is safe during breastfeeding [70].

Regarding the use of VKA during pregnancy, several situations might develop:-In patients requiring long-term treatment with VKA who are attempting pregnancy, it is suggested to substitute LMWH for VKA only when pregnancy is achieved. LMWH should be administered in adjusted doses or 75% of therapeutic doses;-Another option is to switch to UFH or LMWH until the 13th week of gestation, with substitution by VKA until close to delivery when UFH or LMWH is restored;-In pregnant patients with previous VTE, postpartum thromboprophylaxis with prophylactic or intermediate dose of LMWH or VKA targeted at the International Normalized Ratio (INR) 2.0–3.0 is recommended;-In pregnant patients without a history of VTE, homozygous for factor V Leiden or the prothrombin G20210A mutation and with a positive family history for VTE, except antepartum prophylactic- or intermediate-dose LMWH, postpartum thromboprophylaxis with continuing LMWH or VKA targeted at INR 2.0–3.0 is recommended;-In pregnant patients without personal and family history of VTE, homozygous for factor V Leiden or the prothrombin G20210A mutation, postpartum thromboprophylaxis for 6 weeks with prophylactic- or intermediate-dose LMWH or VKA targeted at INR 2.0–3.0 is suggested;-In asymptomatic pregnant patients with all other thrombophilic states and a positive family history for VTE, postpartum thromboprophylaxis with prophylactic or intermediate dose of LMWH or, in those who do not have PC or PS deficiency, VKA targeted at INR 2.0–3.0 is suggested [68].

### 6.6. Direct Oral Anticoagulants

Due to absence of data about the safety and effectiveness of direct oral anticoagulants (DOACs) during pregnancy and breastfeeding, their use in these settings is not currently recommended [70]. A summary of the characteristics of antithrombotic drugs used in pregnancy is available in Table 8 [67,74,76].

## 7. Further Aspects Related to Thromboprophylaxis That Are Useful for Clinical Practice

### 7.1. How to Monitor the Effectiveness of Anticoagulant Thromboprophylaxis

aPTT is the standard coagulation parameter used for the evaluation of the effectiveness of UFH and anti-Xa activity for the monitoring of the safety of LMWH. Along them, point-of-care methods for an assessment of the coagulation status of pregnant patients might also be used (Table 9) [7,77].

According to the RCOG (2015), detection of the peak anti-Xa activity for LMWH administered for treatment of acute VTE during pregnancy or postpartum period is not routinely recommended. Exceptions include patients with extreme body weight (<50 and ≥90 kg) or with other risk factors (renal impairment, recurrency of VTE) (grade of evidence C). The Society of Obstetricians and Gynecologists of Canada (SOGC) (2014) guidelines indicate that its measurement could be considered during the first month of treatment to target levels between 0.6 and 1.0 IU/mL achieved 4 h after the administration of LMWH [77].

### 7.2. How to Manage Cesarean Section in Patients with Anticoagulant Thromboprophylaxis

An overview of the current guidelines providing recommendations about further steps after cesarean delivery is available in Table 10 [77].

### 7.3. How to Manage Neuraxial Anesthesia in Patients Using Anticoagulants

The most dangerous complication of neuraxial anesthesia in patients administered anticoagulation is the development of vertebral canal hematoma. This clinical condition can lead to paralysis when not treated within 12 h [65]. The incidence of spinal epidural hematoma under the setting of neuraxial anesthesia in the obstetric population is rare (1:200,000–1:250,000). In addition to labor pain relief, such an intervention might help in minimizing the need to bear down before complete cervical dilation and contribute to decreasing the levels of circulating catecholamines, which are essential in facilitating vaginal delivery in high-risk settings [7]. Table 11 contains the recommendations associated with the use of NB in the case of anticoagulant treatment [7].

A comparison of the management of NB according to the obstetric and hematologic societies is outlined in Table 12 [67,68,72,77].

Additionally, according to the RCOG, normal risk associated with the insertion of a neuraxial block in pregnant women receiving warfarin occurs in the case of INR ≤ 1.4 [65].

To summarize, the strongest risk factors favoring antenatal thromboprophylaxis are previous recurrent VTE, especially when unprovoked, developed during pregnancy or not associated with surgical intervention, and high-risk thrombophilia (antiphospholipid syndrome, antithrombin, PC or PS deficiency, homozygous form of factor V Leiden mutation, prothrombin mutation or their combination).

However, as outlined above, even in these high-risk clinical situations, different guidelines recommend different time intervals for the beginning of thromboprophylaxis. The SFOG suggests antepartum thromboprophylaxis from the time of detection of pregnancy up to at least six weeks postpartum; the QCG recommends its initiation from the first trimester with similar continuation up to postpartum; the ACCP recommends extension of prophylaxis up to 6 weeks following delivery in the presence of significant prothrombotic risk factors, and the RCOG suggests antenatal thromboprophylaxis in the case of previous VTE, hospital admission, surgery or OHSS.

OHSS as the prothrombotic risk factor is thus the subject of various approaches. According to the QCG, it might be the cause of transient thromboprophylaxis until this state resolves. The SFOG recommends thromboprophylaxis from the time of OHSS development up to the end of postpartum period.

According to the QCG and RCOG, comorbidities including cancer, diabetes mellitus or inflammatory diseases also represent a risk for antenatal thromboprophylaxis. The QCG and SFOG stratify further prenatal and postnatal risk factors and recommend their calculation. Based on the synthesis of these documents, the Asian venous thromboembolism guidelines recommend to patients with four or more risk factors thromboprophylaxis throughout the pregnancy and for six weeks postpartum (for more details, see Table 1) [65,66,67,68].

Regarding the dose of LMWH in various clinical situations, reviewed guidelines recommend different approaches, as well. The ASH and ACCP do not consider an adjusted dose of LMWH. In women receiving anticoagulants before pregnancy, the RANZOG, ASH and ACCP guidelines suggest therapeutic anticoagulation during pregnancy. The ACOG and ACCP guidelines recommend its continuation to the postpartum period, and the RANZOG document prefers the use of pre-pregnancy anticoagulation (Table 2) [70].

The RCOG, ACOG and GTH guidelines consider the heterozygous form of factor V Leiden or prothrombin mutation as the reason for antepartum thromboprophylaxis. Postpartum anticoagulant prophylaxis in these patients is recommended by all guidelines except ASH. In the co-existence of these inherited thrombophilic mutations and in patients with antithrombin deficiency, all guidelines except the ACCP recommend thromboprophylaxis before delivery. In pregnant women with the homozygous form of factor V Leiden mutation and prothrombin variant G20210A, all guidelines suggest anticoagulant prophylaxis throughout the pregnancy and postpartum period, with consideration of further various aspects. In high-risk pregnant patients with antithrombin deficiency, thromboprophylaxis up to six weeks postpartum is recommended.

In patients with PC and PS deficiency, RCOG, ACOG and GTH guidelines suggest prophylaxis with LMWH during pregnancy, and all guidelines recommend thromboprophylaxis in the postpartum period (Table 3) [32,71].

Differences in the recommendations for thromboprophylaxis in patients undergoing cesarean section can also be found. In the case of emergent cesarean delivery, ANZJOG guidelines suggest thromboprophylaxis lasting for 5 days, and RCOG guidelines 10 days, after the delivery. However, further risk factors are considered, as well (Table 10) [77].

Conclusively, most studies consider a previous VTE episode and selected thrombophilia, such as the homozygous form of factor V Leiden mutation and prothrombin variant G20210A, as significant prothrombotic risk factors. Therefore, in these situations, they recommend prolonged thromboprophylaxis during pregnancy and six weeks following delivery. Moreover, there is still general agreement that LMWH is the drug of choice during the pregnancy and postpartum period.

On the other hand, despite the current amount of data on risk factors for prevention of VTE during pregnancy and the postpartum period [70], the available international recommendations are inconclusive regarding antepartum thromboprophylaxis in special situations in patients with thrombophilia. They also differ in the postpartum period, but to a lesser extent [32].

The recommendations for clinical practice are as follows:-To evaluate the need for thromboprophylaxis as soon as the pregnancy is confirmed, consider the presence of the prothrombotic risk factors related to the comorbidities and circumstances of a previous VTE episode;-In patients with previous idiopathic, recurrent or hormone-related VTE, use ante- and also postpartum prophylaxis;-In patients with VTE associated with a major reversible risk factor/without thrombophilic state, use postpartum thromboprophylaxis;-In pregnant patients with prothrombotic risk factors undergoing cesarean section, consider postpartum thromboprophylaxis;-Control changes in the health condition of the patient and modify the actual thromboprophylaxis on an individual basis;-Evaluate the presence of bleeding symptoms and allergic reaction, regularly assess platelet count, function of antithrombin, renal parameters and liver function tests;-In pregnant women with extreme body weight, renal impairment, recurrence of VTE or in the suspicion of noncompliance, evaluate anti-Xa activity of LMWH;-To take control over prothrombotic comorbidities and complications developed during pregnancy, the multidisciplinary approach is preferred;-Along with anticoagulant thromboprophylaxis, use nonpharmacologic prophylaxis, such as intermittent pneumatic compression or elastic stockings.

Limitations of the presented review are the inclusion of studies with different designs—prospective and retrospective studies, randomized controlled trials and also systematic reviews. Moreover, we admit that the review contains data representing a selection bias caused by the inclusion of studies with the absence of a control group. However, these studies confirm the conclusions of the randomized controlled trials and systematic reviews used in this review, as well.

Areas for future research include the relevance of monitoring anti-Xa activity, as well as the impact of the application of risk scores to balance the absolute risk of thromboembolic episodes and bleeding complications. However, currently, when determining the factors increasing the risk of VTE, data are extrapolated mostly from retrospective studies [70,72].

## 8. Conclusions

A previous VTE episode and selected thrombophilia (e.g., the homozygous form of factor V Leiden mutation and prothrombin variant G20210A) are the most significant prothrombotic risk factors. Therefore, in these circumstances, the international guidelines recommend thromboprophylaxis during the pregnancy and postpartum period. LMWH still represents the drug of choice during this period.

The practical implication of this review might be the information about the most appropriate dosing regimens and the summary of the risk of VTE in the course of pregnancy and puerperium in the context of the risk factors discussed.

Data from large prospective registries might provide more information on the absolute risk factors warranting thromboprophylaxis in this setting [70,72].

## Figures and Tables

**Figure 1 pharmaceuticals-17-00773-f001:**
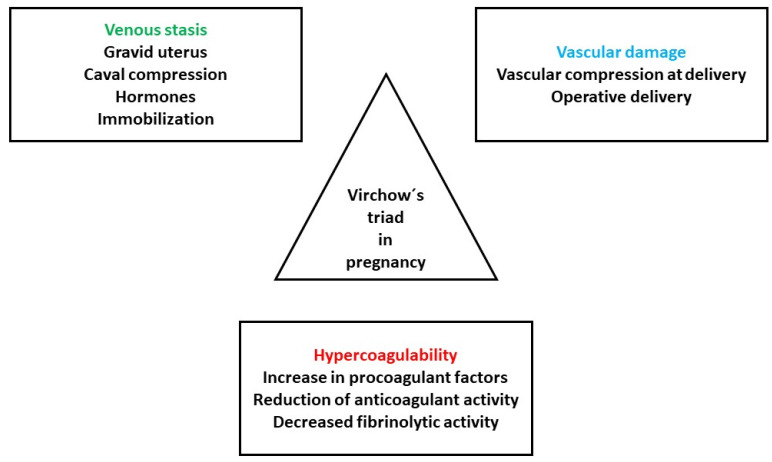
Pathogenesis of VTE in pregnancy. Adapted from [3].

**Figure 2 pharmaceuticals-17-00773-f002:**
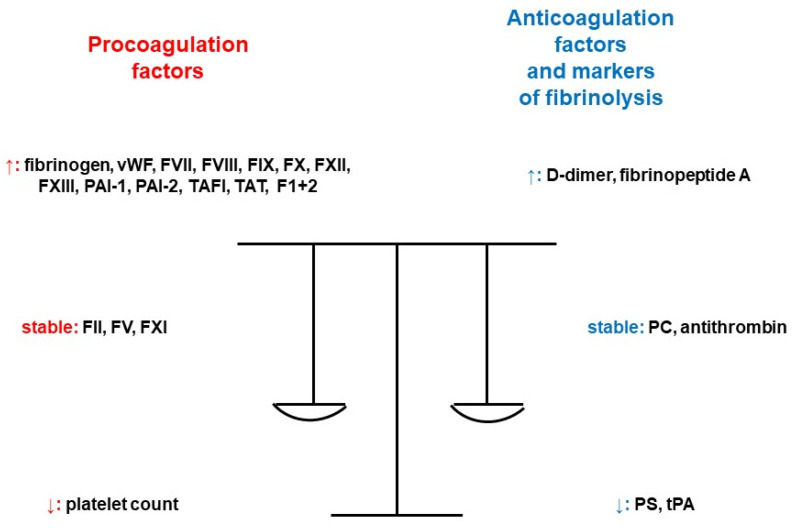
Pregnancy and its impact on the markers of hemostasis [8]. Legend: F1+2; prothrombin fragment 1+2, FII; coagulation factor II, FV; coagulation factor V, FVII; coagulation factor VII, FVIII; coagulation factor VIII, FIX; coagulation factor IX, FX; coagulation factor X, FXI; coagulation factor XI, FXII; coagulation factor XII, FXIII; coagulation factor XIII, PAI-1; plasminogen activator inhibitor-1, PAI-2; plasminogen activator inhibitor 2, PC; protein C, PS; protein S, TAFI; thrombin-activatable fibrinolysis inhibitor, TAT; thrombin–antithrombin complex, tPA; tissue plasminogen activator, vWF; von Willebrand factor (adapted from [8]).

**Figure 3 pharmaceuticals-17-00773-f003:**
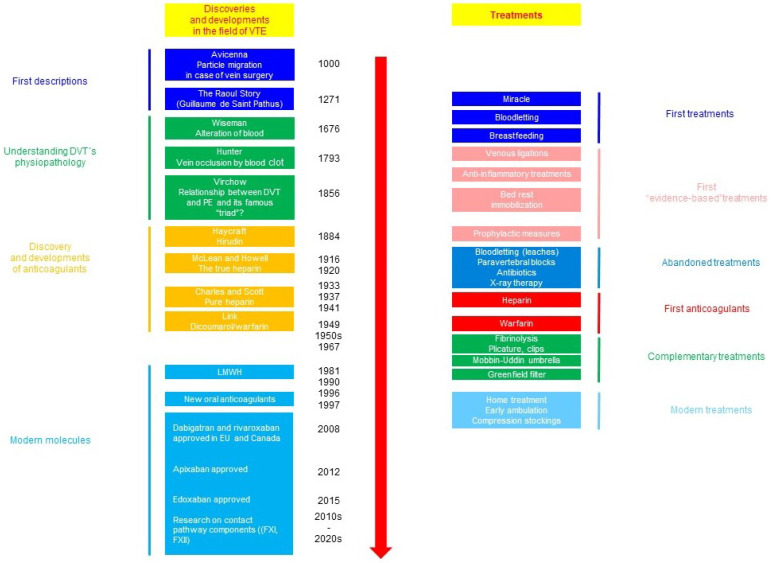
Outline of the history of advances in the management of thromboembolic complications [1,14].

**Table 1 pharmaceuticals-17-00773-t001:** Summary of risk stratification strategies for thromboprophylaxis in pregnancy according to the selected guidelines [65,66,67,68].

Very High Risk Factors	Guideline	Recommendation
Chronic anticoagulation Antithrombin deficiency Repeated episodes of VTE Antiphospholipid syndrome with VTE COVID-19 pneumonia requiring oxygen	*Swedish Society of Obstetrics and Gynecology (SFOG)*	Antepartum thromboprophylaxis from the time of detection of pregnancy up to at least six weeks postpartum
**High Risk Factors**	**Guideline**	**Recommendation**
Previous VTE except an episode associated with major surgery	*Royal College of Obstetricians and Gynecologists (RCOG)*	Antenatal thromboprophylaxis with LMWH
Therapeutic anticoagulation before pregnancy	*Queensland Clinical Guidelines (QCG)*	Antenatal therapeutic anticoagulation with continuation up to 6 weeks after delivery
Previous VTE combined with high-risk thrombophilic state (antiphospholipid syndrome, antithrombin/PC/PS deficiency, homozygous form of factor V Leiden mutation, homozygous form of prothrombin mutation, compound heterozygous form of factor V Leiden and prothrombin mutation) Recurrent unprovoked VTE VTE during pregnancy		
Previous VTE episode not provoked by surgical intervention Recurrent provoked episode of VTE Active inflammatory or autoimmune disorder Comorbidity (cancer, heart failure, nephrotic syndrome, type I diabetes mellitus complicated by nephropathy, sickle cell disease)		Antenatal thromboprophylaxis with LMWH from the first trimester continuing up to postpartum period
Antenatal admission to the hospital OHSS during the first trimester Surgery in the course of pregnancy or postpartum period Severe hyperemesis or dehydration with the need for intravenous fluid		LMWH thromboprophylaxis during stay in the hospital or until this state is resolved
Thrombophilic state with/without family history of VTE		Consideration of thromboprophylaxis during pregnancy and postpartum period
Previous VTE Antiphospholipid syndrome without previous VTE OHSS	*SFOG*	Antepartum thromboprophylaxis from the time of detection of pregnancy up to at least six weeks postpartum
Immobilization (strict bed rest lasting 1 week) Postpartum hemorrhage with blood loss of ≥1 L with the need for surgery Postpartum infection Preeclampsia with IUGR Previous VTE Antithrombin deficiency Factor V Leiden mutation Prothrombin G20210A mutation Systemic lupus erythematosus (SLE) Sickle cell disease Blood transfusion Heart disease	*American College of Chest Physicians (ACCP)*	- State after cesarean section and 1 major or ≥2 minor risk factors, use pharmacologic thromboprophylaxis - If significant risk factors persist after delivery, extend prophylaxis up to 6 weeks following delivery - In patients undergoing cesarean section who are at very high risk for VTE and simultaneously have multiple risk factors that persist in the postpartum period, combine prophylactic LMWH and intermittent pneumatic compression and/or elastic stockings
**Intermediate Risk Factors**	**Guideline**	**Recommendation**
Hospital admission Single prior VTE in relation to major surgery High-risk thrombophilic state (antithrombin, PS or PC deficiency, compound or homozygous state for low-risk thrombophilia) Comorbidities (cancer, active SLE, inflammatory bowel disease, inflammatory polyarthropathy, heart failure, type I diabetes mellitus associated with nephropathy, nephrotic syndrome, sickle cell disease, intravenous use of drugs) Surgical procedure OHSS during the first trimester	*RCOG*	Antepartum thromboprophylaxis with LMWH should be considered
**Minor Risk Factors**	**Guideline**	**Recommendation**
BMI > 30 kg/m^2^ Multiple pregnancies Postpartum hemorrhage > 1 L Smoking > 10 cigarettes/day IUGR Preeclampsia PC deficiency PS deficiency	*ACCP*	Please, see the recommendations for major risk factors for VTE
**Further Risk Factors**	**Guideline**	**Recommendation**
	*QCG*	
*Prenatal risk factors*	*Risk score*	*Recommendation*
Family history of unprovoked/estrogen-associated VTE	1	*Score = 3:* Thromboprophylaxis with LMWH from 28 weeks of gestation *Score ≥ 4:* Thromboprophylaxis with LMWH from the time of evaluation
Episode of VTE provoked by surgical intervention	3	
Age > 35 years	1	
Parity ≥ 1	1	
Smoking		

Gross varicose veins	1	
BMI 30−39 kg/m^2^	1	
BMI ≥ 40 kg/m^2^	2	
IVF/assisted reproductive technology	1	
Multiple pregnancies	1	
Preeclampsia	1	
Immobility	1	
Systemic infection	1	
Diabetes mellitus	1	
*Postnatal risk factors*	*Risk score*	*Recommendation*
		Postnatal risk score = antenatal risk factors score + postnatal risk factors score *2 points:* Thromboprophylaxis until discharge *≥3 points:* thromboprophylaxis lasting 7 days/longer if ongoing risk
Cesarean section	3	
Elective cesarean section	1	
Labor lasting > 24 h	1	
Operative vaginal birth	1	
Preterm birth	1	
Peripartum hemorrhage > 1 L or requiring transfusion	1	
Stillbirth	1	
Cesarean hysterectomy	3	
	*SFOG*	
*Risk factor*	*Risk score*	*Recommendation*
Heterozygous factor V Leiden	1	*1 point:* lifestyle advice *2 points:* postpartum thromboprophylaxis at least 7 days and short-term, prophylaxis during the presence of temporary risk factor *3 points:* postpartum thromboprophylaxis for at least six weeks
Heterozygous prothrombin mutation	1	
Age > 40 years	1	
BMI 30–40	1	
VTE episode among first-grade relatives aged < 50 years		
Inflammatory bowel disease	1	
Homocysteine > 8 mmol/L in the course of pregnancy	1	
Comorbidities (cancer and its treatment, SLE, heart disease, sickle cell disease, essential thrombocytosis)	1	
PS deficiency	2	
PC deficiency	2	
Fracture cast, immobilization due to strict bed rest or COVID-19 infection	2	
BMI > 40	2	
Homozygous factor V Leiden mutation	3	
Homozygous prothrombin mutation	3	
Double mutation	3	
*Postnatal risk factors*	*Risk score*	
Preeclampsia	1	
Placental abruption	1	
Cesarean section	1	
Blood transfusion	1	
Stillbirth	1	
Severe infection/sepsis	1	

Legend: BMI—body mass index, COVID-19—Coronavirus Disease 19, IUGR—intrauterine growth restriction, IVF—in vitro fertilization, LMWH—low-molecular-weight heparin, OHSS—ovarian hyperstimulation syndrome, PC—protein C, PS—protein S, RCOG—Royal College of Obstetricians and Gynecologists, VTE—venous thromboembolism.

**Table 2 pharmaceuticals-17-00773-t002:** Summary of recommendations regarding the circumstances of previous VTE in pregnant patients [70].

Situation	American College of Obstetricians and Gynecologists (ACOG)	ACCP	American Society of Hematology (ASH)	The Royal Australian and New Zealand College of Obstetricians and Gynecologists (RANZOG)	Society of Obstetricians and Gynecologists of Canada (SOGC)
**History of idiopathic** **VTE or VTE with hormones (estrogen)**	Pharmacologic thromboprophylaxis of VTE during pregnancy and postpartum period	Pharmacologic thromboprophylaxis of VTE during pregnancy and postpartum period	Pharmacologic thromboprophylaxis of VTE during pregnancy and postpartum period	Pharmacologic thromboprophylaxis of VTE during pregnancy and postpartum period	Pharmacologic thromboprophylaxis of VTE during pregnancy and postpartum period
**Previous VTE associated with a major reversible risk factor/** **without thrombophilic state**	*Postpartum period:* pharmacologic thromboprophylaxis in the presence of additional risk factors (cesarean section family history, etc.)	*Postpartum period:* pharmacologic thromboprophylaxis	*Postpartum period:* pharmacologic thromboprophylaxis	*Postpartum period:* pharmacologic thromboprophylaxis	
**Previous recurrent VTE**	Pharmacologic thromboprophylaxis during pregnancy and postpartum period	Pharmacologic thromboprophylaxis during pregnancy and postpartum period		Pharmacologic thromboprophylaxis during pregnancy and postpartum period	Pharmacologic thromboprophylaxis during pregnancy and postpartum period
**Woman** **receiving** **anticoagulants** **who becomes** **pregnant**	*Pregnancy:* adjusted dose of LMWH/UFH *Postpartum period:* therapeutic anticoagulation with VKA/ LMWH	*Pregnancy:* LMWH in therapeutic dose or 75% of the dose *Postpartum period:* therapeutic anticoagulation with VKA/ LMWH	*Pregnancy:* LMWH in one dose or twice a day *Postpartum period:* UFH/LMWH/fondaparinux fondaparinux not recommended during breastfeeding (2C), VKA	*Pregnancy:* Therapeutic anticoagulation *Postpartum period:* Return to prepregnancy anticoagulation	
**Doses** **of LMWH**	Prophylactic, intermediate or adjusted dose during pregnancy and postpartum period	Prophylactic or intermediate dose during pregnancy and postpartum period	*Pregnancy:* Standard dose *Postpartum period:* Standard or intermediate dose	Prophylactic, intermediate or adjusted dose during pregnancy and postpartum period	Prophylactic, intermediate or adjusted dose during pregnancy and postpartum period

Legend: ACCP—American College of Chest Physicians, LMWH—low-molecular-weight heparin, UFH—unfractionated heparin, VKA—vitamin K antagonists, VTE—venous thromboembolism.

**Table 3 pharmaceuticals-17-00773-t003:** Summary of recommendations for thromboprophylaxis in pregnancy according to the presence of thrombophilic state [32,71].

	ASH	SOGC	RCOG	ACOG	ACCP	Society of Thrombosis and Haemostasis Research/Gesellschaft für Thrombose und Hämostaseforschung (GTH)
**Heterozygous form of factor V Leiden or prothrombin mutation**						
*Antepartum*	No	No	Yes, if additional risk factors from * with a total score of 3 are present—thromboprophylaxis throughout the pregnancy; if further risk factors from * with a total score of 2 are present, thromboprophylaxis from 28 weeks of gestation should be evaluated (D)	Surveillance or prophylactic LMWH/UFH	No (2C)	+/−
*Postpartum*	No	Thromboprophylaxis in the presence of 2 risk factors: BMI ≥ 30 kg/m^2^, (II-2B), smoking > 10 cigarettes/day (II-2B), preeclampsia (II-2B), preterm delivery (III-B), IUGR (II-2B), placenta previa (II-2B), emergency cesarean section (II-2B), blood loss of >1 L or need for transfusion (II-2B), stillbirth (III-B), comorbidity (cardiac disease, varicose veins, SLE, inflammatory disease, sickle cell disease, gestational diabetes mellitus (III-B). The length of such thromboprophylaxis should be 6 weeks (II-3B)	Thromboprophylaxis for at least 10 days after delivery if additional risk factor from * with a total score of 1 is present; If a woman has a positive family history of VTE, thromboprophylaxis ought to last for six weeks (D)	Surveillance or anticoagulation if there are further risk factors (first-degree relative with thromboembolic episode aged <50 years, obesity, prolonged immobilization)	Yes, if there is positive family history of VTE with prophylactic or intermediate dose of LMWH/ VKA) targeted at International Normalized Ratio (INR) 2–3 (2C)	+/− in the case of negative family history, otherwise yes
**PC/PS deficiency**						
*Antepartum*	No	No prophylaxis with LMWH should be evaluated (D)	Thromboprophylactic LMWH/UFH	Surveillance	No (2C)	Yes
*Postpartum*	Yes, for women with a family history of VTE	Surveillance or thromboprophylaxis in the presence of 2 risk factors: BMI ≥ 30 kg/m^2^ (II-2B), smoking > 10 cigarettes/day (II-2B), preeclampsia 838 (II-2B), IUGR (II-2B), placenta previa (II-2B), emergency cesarean section (II-2B), blood loss of >1 L or need for transfusion (II-2B), preterm delivery (III-B), stillbirth (III-B), comorbidity cardiac disease, varicose veins, SLE, inflammatory disease, sickle cell disease, gestational diabetes mellitus (III-B). The length of such thromboprophylaxis should be 6 weeks (II-3B)	Yes with LMWH (D)	Surveillance or anticoagulation in the presence of further risk factors (first-degree relative with episode of VTE aged <50 years obesity, prolonged immobilization)	Prophylactic or intermediate dose of LMWH if a family history of VTE is positive (2C)	Yes
**Compound** **heterozygosity**						
*Antepartum*	Yes	Yes with LMWH (IIIB) should be evaluated (D)	Thrombo-prophylaxis with LMWH	Prophylactic LMWH/UFH	No (2C)	Yes
*Postpartum*	Yes	Yes with LMWH (II-3B)	Yes with LMWH (D)	Anticoagulation	Yes, if there is positive family history of VTE with prophylactic or intermediate dose of LMWH/ VKA targeted at INR 2–3 (2C)	Yes
**Homozygous form of factor V** **Leiden or** **prothrombin** **mutation**						
*Antepartum*	Yes for factor V Leiden mutation, for the prothrombin mutation only in the case of positive family history of VTE	Yes, LMWH (II-2A for factor V Leiden mutation, IIIB for prothrombin G20210A mutation)	Thromboprophylaxis with LMWH should be evaluated (D)	Prophylactic LMWH/ UFH	Prophylactic or intermediate dose of LMWH if a family history of VTE is positive (2B)	Yes
*Postpartum*	Yes	Yes, LMWH (II-2B) with the length of thromboprophylaxis 6 weeks (II-3B)	Yes, LMWH (D)	Anticoagulation	Yes, with prophylactic or intermediate dose of LMWH/ VKA targeted at INR 2–3 (2B)	Yes
**Antithrombin** **deficiency**						
*Antepartum*	Yes, in the case of positive family history of VTE	Yes, LMWH (IIIB)	Yes, from at least 28 weeks of gestation; if further risk factor from those in * with a total score of 1 is present, beginning from the first trimester (D)	Prophylactic LMWH/UFH	No (2C)	Yes
*Postpartum*	Yes in the case of positive family history of VTE	Yes, LMWH (II-2B) lasting for 6 weeks after delivery (II-3B)	Yes, LMWH (D)	Anticoagulation	Yes with prophylactic or intermediate dose of LMWH/ VKA targeted at INR 2–3 (2C)	Yes

* Factors to calculate the total score: for 1 point: immobilization and/or dehydration, preeclampsia, systemic infection, multiple pregnancies, assisted reproduction technology, midcavity or rotational operative delivery, elective cesarean section, postpartum hemorrhage (>1 L/the need for transfusion), prolonged labor (>24 h), preterm birth ≤ 37 weeks of gestation, smoking, parity ≥ 3, gross varicose veins, obesity (BMI ≥ 30), age > 35 years, positive family history of unprovoked or estrogen-associated VTE in first-degree relative; for 2 points: cesarean delivery, obesity with BMI ≥ 40; for 3 points: hyperemesis, surgery except immediate reconstruction of the perineum or comorbidity (cancer, active SLE, heart failure, inflammatory bowel disease or inflammatory polyarthropathy, sickle cell disease, type I diabetes with nephropathy, nephrotic syndrome, intravenous drug use); for 4 points: OHSS during the first trimester. Legend: ACCP—American College of Chest Physicians, ACOG—American College of Obstetricians and Gynecologists, ASH—American Society of Hematology, BMI—body mass index, LMWH—low-molecular-weight heparin, OHSS—ovarian hyperstimulation syndrome, PC—protein C, PS—protein S, RCOG—Royal College of Obstetricians and Gynecologists, SLE—systemic lupus erythematosus, SOGC—Society of Obstetricians and Gynecologists of Canada, SLE—systemic lupus erythematosus, UFH—unfractionated heparin, VKA—vitamin K antagonists, VTE—venous thromboembolism.

**Table 4 pharmaceuticals-17-00773-t004:** Aspects of pharmacologic thromboprophylaxis for further consideration [67].

Aspect	Consideration
**Context**	- Limited evidence for thromboprophylaxis in the course of pregnancy and postpartum period - Need for individualization according to the presence of risk factors and clinical state - Various evidence regarding ↓ creatinine clearance and potential LMWH accumulation
**Animal origin** **of heparin**	- Heparin sodium is derived from porcine (fondaparinux is a synthetic anticoagulant that might be an alternative when LMWH is unsuitable)
**Contraindication**	- Hypersensitivity reaction - Heparin-induced thrombocytopenia (HIT) - Creatinine clearance < 15 mL/minute *additional specific contraindications:* *enoxaparin:* - Active major bleeding - Hypersensitivity to benzyl alcohol *dalteparin:* - Active major bleeding - Neuraxial blockade (NB) *tinzaparin:* - Hypersensitivity to benzyl alcohol or sodium metabisulphite - Active major bleeding or conditions representing an ↑ risk of hemorrhage - Uncontrolled severe arterial hypertension - Hemorrhagic or diabetic retinopathy
**Caution**	- Renal impairment (creatinine clearance < 30 mL/minute) (unfractionated heparin (UFH) might be more suitable) - Abnormal liver function tests - Thrombocytopenia (platelet count < 100 × 10^9^/L) (↑ bleeding risk, but not ↓ the risk of VTE)
**Risk factors of bleeding**	- Active bleeding (requiring transfusion of at least 2 units of blood products during 24 h or primary postpartum hemorrhage > 1 L) - Chronic and clinically significant bleeding lasting > 48 h - ↑ risk of major hemorrhage (e.g., placenta previa) - Bleeding disorders - Recent bleeding in the central nervous system - Intracranial/spinal lesion - Abnormal coagulation parameters - Thrombocytopenia - Severe platelet dysfunction or antiplatelet treatment - Active peptic ulcer - Obstructive jaundice (cholestasis) - Recent major surgery with ↑ bleeding risk - Use of drugs affecting clotting - NB or lumbar puncture

Legend: LMWH—low-molecular-weight heparin, VTE—venous thromboembolism.

**Table 5 pharmaceuticals-17-00773-t005:** Dosing regimens of UFH [70].

Regimen	Dosage
**Prophylactic**	*1st trimester:* 5000–7500 IU subcutaneously (SC) twice daily *2nd trimester:* 7500–10,000 IU SC twice daily *3rd trimester:* 10,000 IU twice daily
**Adjusted (according to aPTT)**	10,000 IU or more, SC twice daily—adjusted for aPTT to be between 1.5–2.5 times control 6 h after administration

Legend: aPTT—activated partial thromboplastin time, UFH—unfractionated heparin.

**Table 6 pharmaceuticals-17-00773-t006:** Dosing regimens of UFH according to the weight of the pregnant patient [67].

Weight (kg)	Standard Prophylactic Dosage (IU SC)
<50 50–90 91–130 131–170 >170	Consider reduced dosage 5000 twice a day 7500 twice a day
**Weight (kg)**	**High prophylactic dosage (IU) ***
<50 50–130 ≥130	Consider reduced dosage 7500 twice a day 7500 thrice a day

* High prophylactic dose is recommended for patients with several significant risk factors (previous DVT during standard prophylactic dose, antiphospholipid syndrome and previous DVT, increased risk of arterial thrombosis). Legend: DVT—deep venous thrombosis, IU—international unit, SC—subcutaneously, UFH—unfractionated heparin.

**Table 7 pharmaceuticals-17-00773-t007:** Dosages of LMWHs [65,66,67,70,72,73,74].

Enoxaparin						
Weight (kg)	Dosage (RCOG)	Dosage (SFOG)	Dosage (SOGC)	Dosage (QCG)	Dosage (ASH, ACCP)	Dosage (ACOG)
<50 kg	20 mg/day	*Very high risk of VTE (measurable anti-Xa activity before the next dose (≥0.05–0.1 IU/mL)):* 20 mg twice a day	*Prophylactic dose:* 40 mg/day (30 mg twice a day) *- obese patients:* 60 mg/day			40 mg/day
50–90	40 mg/day	*Very high risk of VTE:* 40 mg twice a day		40 mg/day	40 mg/day	
91–130	60 mg/day	*Very high risk of VTE:* 60 mg twice a day		60 mg/day		
131–170	80 mg/day			80 mg/day		
>170	0.6 mg/kg/day			0.5 mg/kg/day		
*High prophylactic dose for patients weighing 50–90 kg*	40 mg twice a day		*Intermediate dose *:* 40 mg twice a day	*High prophylactic* *dose for patients weighing < 50 kg%:* 40 mg/day *High prophylactic* *dose for patients* *weighing 50–130 kg%:* 80 mg/day *High prophylactic* *dose for patients weighing > 131 kg%:* 60 mg twice a day	*Intermediate dose ^&^:* 40 mg twice a day or 80 mg/day *Weight-adjusted dose (ACCP):* 1 mg/kg twice a day	*Intermediate dose+:* 40 mg twice a day
**Dalteparin**						
**Weight (kg)**	**Dosage** **(RCOG)**	**Dosage** **(SFOG)**	**Dosage** **(SOGC)**	**Dosage** **(QCG)**	**Dosage** **(ASH, ACCP)**	**Dosage** **(ACOG)**
<50	2500 IU/day	*Very high risk of VTE:* 2500 IU twice a day	*Prophylactic dose:* 5000 IU/day *- >20 weeks of gestation:* 5000 IU twice a day *- obesity:* 7500 IU/day	2500 IU/day		5000 IU/day
50–90	5000 IU/day	*Very high* *risk of VTE:* 5000 IU twice a day		5000 IU/day	5000 IU/day	
91–130	7500 IU/day	*Very high risk of VTE:* 7500 IU twice a day		7500 IU/day		
131–170	10,000 IU/day			10,000 IU/day		
>170	75 IU/kg/day			75 IU/kg/day		
High prophylactic dose for patients weighing 50–90 kg	5000 IU twice a day		*Intermediate dose *:* 100 IU/kg/day or 5000 IU twice a day	*High prophylactic* *dose* *for patients* *weighing* *<50 kg%:* 2500 IU twice a day *High prophylactic dose for patients weighing 50–130 kg%:* 5000 IU twice a day *High prophylactic dose for patients weighing >131 kg%:* 7500 IU twice a day	*Intermediate dose ^&^:* 5000 IU twice a day or 10,000 IU/day	*Intermediate* *dose+:* 5000 IU twice a day
		*<90 kg:* 5000 IU/day			*Weight-adjusted dose (ACCP):* 200 IU/kg or 100 IU/kg twice a day	
		*>90 kg:* 7500 IU/day				
**Tinzaparin**						
**Weight (kg)**	**Dosage** **(RCOG)**	**Dosage** **(SFOG)**	**Dosage** **(SOGC)**	**Dosage** **(QCG)**	**Dosage** **(ASH, ACCP)**	**Dosage** **(ACOG)**
<50	3500 IU/day	*Very high risk of VTE:* 2500 IU twice a day	4500 IU/day *- obesity:* 75 IU/kg/ day			
50–90	4500 IU/day	*Very high risk of VTE:* 4500 IU twice a day			4500 IU/day	
91–130	7000 IU/day	*Very high risk of VTE:* 4500 + 8000 IU/day			75 IU/kg/day in the case of extreme body weight	
131–170	9000 IU/day					
>170	75 IU/kg/day					
			*Intermediate dose *:* 4500 IU twice a day or 9000 IU/day		*Intermediate dose ^&^:* 10,000 IU/day	
		*<90 kg:* 4500 IU/day			*Weight-adjusted dose (ACCP):* 175 IU/kg/day	
		*>90 kg:* 8000 IU/day				
**Nadroparin**						
**Weight (kg)**	**Dosage** **(RCOG)**	**Dosage** **(SFOG)**	**Dosage** **(SOGC)**	**Dosage** **(QCG)**	**Dosage** **(ASH, ACCP)**	**Dosage** **(ACOG)**
			*Prophylactic dose:* 2850 IU/day		*Prophylactic dose:* 2850 IU/day	

*—indicated in women with a previous VTE episode and a high-risk thrombophilic state (antiphospholipid syndrome, antithrombin deficiency) not previously on anticoagulation medication. +—ACCP defined intermediate dose as following: enoxaparin 40 mg twice a day or dalteparin 5000 IU twice a day. ACOG recommends intermediate dosage, as the prophylactic LMWH dose increased with the progression of the pregnancy and the weight of patients up to a maximum dosage of enoxaparin of 1 mg/kg/day [75]. %—high prophylactic dose should be considered in patients with several significant risk factors of VTE (DVT during treatment with standard prophylactic dose, DVT and antiphospholipid syndrome and in women with an increased risk of arterial thrombosis (hyperhomocysteinemia)). &—<75% of a therapeutic dose and higher than prophylactic dose, higher dose might be administered with increased weight of the pregnant patient (LMWH is adjusted to a peak anti-Xa activity 0.2–0.6 IU/mL) [73]. Legend: ACCP—American College of Chest Physicians, ACOG—American College of Obstetricians and Gynecologists, ASH—American Society of Hematology, DVT—deep venous thrombosis, IU—international unit, QCG—Queensland Clinical Guidelines, RCOG—Royal College of Obstetricians and Gynecologists, SFOG—Swedish Society of Obstetrics and Gynecology, SOGC—Society of Obstetricians and Gynecologists of Canada, VTE—venous thromboembolism.

**Table 8 pharmaceuticals-17-00773-t008:** The most important features of antithrombotic drugs used in pregnancy and during postpartum period [67,74,76].

Drug	Aspect	Consideration
**LMWH**	**Fetus**	- Does not cross placenta - No teratogenicity and advanced risk of fetal bleeding - Safe during breastfeeding (detectable low levels, but not orally absorbed)
	**Safety**	- Thrombocytopenia 0.08% - Allergic skin reaction 1.84% - LMWH is preferred to UFH - due to: - ↓ incidence of bleeding - ↓ risk of HIT - ↓ risk of osteoporosis
	**Monitoring**	- Baseline platelet count - Serum creatinine
	**Recommendation**	- Drug of choice for ante- and postnatal prophylaxis - ↓ dose in the case of renal impairment or use UFH - ↑ dose when detecting antithrombin deficiency
**UFH**	**Fetus**	- Does not cross the placenta - No teratogenicity - Safe during breastfeeding
	**Safety**	- ↑ bruising after the administration - Might be preferred: - in the case of significant renal dysfunction - with the need for rapid reversal - in patients with high risk of VTE and need of neuraxial blockade
	**Monitoring**	- Platelet count - Development of HIT
	**Recommendation**	- Not the first choice in pregnancy - Switch from LMWH to UFH before the onset of labor
**Warfarin**	**Consideration**	- Crosses placenta and might lead to fetal hemorrhage - Safe during breastfeeding - In women with mechanic heart valves - After delivery if prolonged thromboprophylaxis is indicated
**Fondaparinux**	**Consideration**	- In women with severe allergic reaction to heparin who cannot use danaparoid - Pause 5 days before delivery (long half-life)
**Danaparoid**	**Consideration**	- Undetectable in breast milk
**DOACs**	**Consideration**	- Not recommended during pregnancy, breastfeeding (detectable—low levels present in the breast milk in the case of rivaroxaban) and with the need for NB - Passes the placenta - Can lead to congenital abnormalities considered as embryopathy
**Acetylsalicylic acid (ASA)**	**Consideration**	- Not recommended as the single drug for VTE prophylaxis - No adverse effect of low-dose ASA for the prevention of preeclampsia

Legend: DOACs—direct oral anticoagulants, HIT—heparin-induced thrombocytopenia, LMWH—low-molecular-weight heparin, NB—neuraxial blockade, UFH—unfractionated heparin, VTE—venous thromboembolism.

**Table 9 pharmaceuticals-17-00773-t009:** Advantages and disadvantages of the use of various laboratory methods in pregnancy [7,77].

Method	Application	Limitation
**aPTT**	- To modify high-dose UFH therapy to target range - To inform about the safe timing of NB	- ↓ and variable sensitivity for LMWH - Normal range is different in pregnancy and between laboratory reagents - In term pregnancy, its response to UFH is ↓ because of ↑ fibrinogen level and FVIII activity and due to ↑ nonspecific protein binding
**anti-Xa activity**	- To adjust high-dose treatment with LMWH or UFH to target range - Undetectable level (<0.01 IU/mL) might be reassuring for NB - Suggests little to absent residual effect of LMWH or UFH	- It does not represent the effect of antithrombin on UFH - Particular LMWHs have various affinities for anti-Xa - Might not correlate with actual effect of agent under the conditions in vivo - Describes pharmacokinetics, but not pharmacodynamics—test might not be rapidly available - Threshold for safe NB and other clinical states is unknown
**Point-of-care:** **Thromboelastography (TEG) and rotational thromboelastometry (ROTEM)**	- Data about clot formation might be available up to 15–20 min - Normal ranges for TEG were established - Occasional case reports with their use in pregnant women administered thromboprophylaxis to evaluate suitability for NB have been published	- Safe reference ranges for NB in pregnant women using thromboprophylaxis have not been established - Currently, they cannot be used for the safe evaluation of the possibility of pregnant women to undergo NB

Legend: aPTT—activated partial thromboplastin time, FVIII—coagulation factor VIII, LMWH—low-molecular-weight heparin, NB—neuraxial blockade, UFH—unfractionated heparin.

**Table 10 pharmaceuticals-17-00773-t010:** Recommendations regarding thromboprophylaxis with LMWH after cesarean delivery [77].

Feature	ASH (2018)	RCOG (2015)	SOGC (2014)	ACCP (2012)	ANZJOG (2012)
**Elective** **cesarean** **delivery**	In patients with no or 1 risk factor (excluding thrombophilic state or previous VTE), no antepartum and postpartum, thromboprophylaxis is suggested	no	no	no (1B)	no
**Emergent** **cesarean** **delivery**		All patients who had cesarean section ought to be considered (C) for thromboprophylaxis with LMWH lasting for 10 days after delivery except women undergoing an elective cesarean delivery, who ought to be considered for prophylaxis for 10 days following delivery if they have any further risk factors (C)	no		Thromboprophylaxis with LMWH/ UFH for ≥5 days until recovery of full mobility (evidence level 1 with agreement of all authors of the guideline)
**Elective** **cesarean** **delivery and risk factors**	Not stated	See above (consider prophylaxis with LMWH for 10 days following delivery) (C)	Postpartum prophylaxis ought to be considered in the presence of 2 risk factors (emergency cesarean section is 1 risk factor) (II-2B), LMWH up to 2 weeks if 2 risk factors are present *	In patients with 1 major or ≥2 minor risk factors, prophylactic LMWH or mechanic prophylaxis in the case of contraindication to anticoagulants during stay in the hospital is recommended (2B) ^&^	*≥1 major and ≥2 minor risk factors:* thromboprophylaxis for ≥5 days or until fully mobile @ *1 major and 2 minor risk factors:* consider graduated compression stockings@
**Emergent** **cesarean** **delivery and risk factors**	Not stated	See above (C)	Postpartum prophylaxis in the presence of any 3 or more risk factors (elective cesarean section is 1) (II-2B), LMWH up to 2 weeks if 1 risk factor is present *	Presence of at least 1 major risk factor or at least 2 minor risk factors (planned cesarean section) or 1 minor risk factor in the case of an emergency cesarean delivery ^&^	Thromboprophylaxis with LMWH or UFH for at least 5 days or longer until restoration of full mobility (group consensus of all authors, level of evidence 1) @

&—major risk factors: immobilization, postpartum hemorrhage ≥ 1 L with the need for surgery, history of VTE, preeclampsia with IUGR, thrombophilia (antithrombin deficiency, factor V Leiden mutation, prothrombin G20210A mutation), comorbidities (SLE, sickle cell disease, heart disease), postpartum infection, blood transfusion; minor risk factors: BMI > 30 kg/m^2^, multiple pregnancies, postpartum hemorrhage > 1 L, smoking > 10 cigarettes/day, IUGR, thrombophilia (PC/PS deficiency), preeclampsia. *—any 2 of the following risk factors: BMI ≥ 30 kg/m^2^, smoking > 10 cigarettes/day, preeclampsia, IUGR, placenta previa, emergency cesarean delivery, peri- or postpartum blood loss of > 1 L or transfusion administration, low-risk thrombophilia (PC/PS deficiency), heterozygous form of factor V Leiden mutation or prothrombin G20210A mutation, heart disease, SLE, inflammatory bowel disease, sickle cell disease, varicose veins, gestational diabetes mellitus, stillbirth, preterm delivery; any 3 or more of the following risk factors: age > 35 years, parity ≥ 2, multiple pregnancies, assisted reproductive technology, premature rupture of membranes, placental abruption, elective cesarean delivery, maternal cancer. @—major risk factors: elective cesarean section, BMI ≥ 30 kg/m^2^, immobilization, comorbidity (SLE, inflammatory bowel disease, pneumonia), systemic infection, preeclampsia; minor risk factors: age > 35 years, prolonged labor (>24 h), smoking, postpartum hemorrhage > 1 L, gross varicose veins, extensive perineal trauma and its prolonged healing. Legend: ACCP—American College of Chest Physicians, ASH—American Society of Hematology, ANZJOG—The Australian and New Zealand Journal of Obstetrics and Gynecology, BMI—body mass index, LMWH—low-molecular-weight heparin, PC—protein C, PS—protein S, RCOG—Royal College of Obstetricians and Gynecologists, SCOG—Society of Obstetricians and Gynecologists of Canada, SLE—systemic lupus erythematosus, UFH—unfractionated heparin, VTE—venous thromboembolism.

**Table 11 pharmaceuticals-17-00773-t011:** Management of NB in pregnant patients using anticoagulants according to the Society for Obstetric Anesthesia and Perinatology [7].

Drug	Intrapartum		Postpartum
	*Elective*	*Urgent* *and emergent*	
**Subcutaneous UFH**	*Low-dose UFH thromboprophylaxis 5000 IU twice a day or 5000 IU thrice a day:* pause for 4–6 h before NB or assessment of coagulation status (IIa C-EO) *Intermediate-dose UFH thromboprophylaxis (7500 IU twice a day or 10,000 IU twice a day):* pause 12 h and assessment of coagulation parameters before NB (IIa C-EO) *High-dose UFH (individual dose >10,000 IU or >20,000 IU daily dose):* pause 24 h before NB and assessment of coagulation parameters (IIa C-EO)	*Low-dose UFH thromboprophylaxis:* pause 4–6 h following the last dose before NB or assessment of coagulation parameters; in urgent situation, with greater risk of general anesthesia in the comparison with NB, neuraxial procedure is possible (IIa C-EO) *Intermediate-dose UFH thromboprophylaxis:* pause 12 h following the last dose before NB and assessment of coagulation status - In urgent situation, neuraxial procedure rather than general anesthesia is preferred (IIa C-EO) *High-dose UFH:* if it is ≥24 h since the last dose, and normal coagulation parameters were obtained (normal aPTT or undetectable antiXa activity), NB is possible (IIb C-EO)	*UFH thromboprophylaxis:* pause ≥ 1 h after NB and after catheter removal before UFH administration - Indwelling catheters might be maintained with low dose of UFH (5000 IU twice a day) - Pause non-steroidal anti-inflammatory drugs, not acetaminophen until catheter removal, if using thromboprophylaxis
**Intravenous UFH**	Stop infusion for 4–6 h and then assess coagulation status before NB (IIa C-EO)		Pause ≥ 1 h after NB before restarting anticoagulation (IIb C-EO)
**LMWH**	*Low-dose LMWH thromboprophylaxis (enoxaparin ≤ 40 mg once a day or 30 mg twice a day, or dalteparin 5000 IU once a day):* pause ≥ 12 h before placing NB (I C-EO) *Intermediate-dose LMWH thromboprophylaxis (enoxaparin > 40 mg once a day or 30 mg twice a day and <1 mg/kg twice a day or 1.5 mg/kg once a day or dalteparin > 5000 IU once a day and <120 IU/kg twice a day or 200 IU/kg once a day):* potential pause 12–24 h before NB (IIb C-EO) *Higher dose of LMWH (enoxaparin 1 mg/kg twice a day or 1.5 mg/kg once a day; dalteparin 120 IU/kg twice a day or 200 IU/kg once a day):* pause ≥ 24 h before NB (I C-EO)	*Low-dose LMWH thromboprophylaxis* if pause ≥ 12 h, low risk of NB (I C-EO); if pause < 12 h before NB, insufficient data to recommend NB (IIb C-EO) - In high-risk situation, risk of general anesthesia is greater than NB (IIb C-EO) *Intermediate-dose LMWH thromboprophylaxis:* insufficient data to specify pause 12–24 h before NB (IIb C-EO) *Higher dose of LMWH:* if pause ≥ 24 h, low risk for NB (I C-EO); if pause < 24 h, insufficient data to recommend NB (IIb C-EO)	*Low-dose LMWH thromboprophylaxis* pause ≥ 12 h after NB and ≥4 h after catheter removal before restarting LMWH - Indwelling catheters might be maintained with low dose of LMWH: - Removal can take place ≥ 12 h after a last LMWH dose, and next dose of LMWH can be administered in ≥4 h (I C-EO) - Pause non-steroidal anti-inflammatory drugs, not acetaminophen, until removal if using thromboprophylaxis (IIa C-EO) *Higher dose of LMWH:* pause ≥ 24 h after NB and ≥4 h after catheter removal before restarting LMWH thromboprophylaxis (I C-EO)

Legend: IU—international unit, LMWH—low-molecular-weight heparin, NB—neuraxial blockade, UFH—unfractionated heparin.

**Table 12 pharmaceuticals-17-00773-t012:** NB in relation to anticoagulant thromboprophylaxis in pregnancy according to further international guidelines [67,68,72,77].

	SOGC	QCG	ACCP	ANZJOG
Last dose to neuraxial block	Hours to wait			
**LMWH**	10–12	12	24 (1B)	12
**UFH**	Maximally 10,000 IU/day; no pause unless abnormal coagulation parameters are obtained	4 (not in the case of dosages >5000 IU twice or thrice a day)	24 in patients using adjusted dose of subcutaneous UFH (twice daily) (1B)	
**Neuraxial block to next dose**	**Hours to wait**			
**LMWH**	6–8 >24 if bleeding during NB	4		
**UFH**	1–8	1 (not in the case of dosages >5000 IU twice or thrice a day)		
**Last dose to catheter removal**	**Hours to wait**			
**LMWH**	10–12	12		12
**UFH**	4	4 (not in the case of dosages >5000 IU twice or three times a day)		
**Catheter removal to the next dose**	**Hours to wait**			
**LMWH**	>4	4		
**UFH**		1 (not in the case of dosages >5000 IU twice or three times a day)		

Legend: ACCP—American College of Chest Physicians, ANZJOG—The Australian and New Zealand Journal of Obstetrics and Gynecology, IU—international unit, LMWH—low-molecular-weight heparin, NB—neuraxial blockade, QCG—Queensland Clinical Guidelines, SCOG—Society of Obstetricians and Gynecologists of Canada, UFH—unfractionated heparin.

## Data Availability

Data sharing not applicable.

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
