# Peer review of "Evaluating Thromboprophylaxis Strategies for High-Risk Pregnancy: A Current Perspective"

_pharmaceuticals, 2024, doi:10.3390/ph17060773_

Round 1

Reviewer 1 Report

Comments and Suggestions for Authors

Review Report on the Manuscript "Current approaches for thromboprophylaxis in pregnancy with high thromboembolism risk"

The review article aims to address the crucial issue of venous thromboembolism (VTE) during pregnancy, identifying the condition as a leading cause of maternal death in developed countries. the following suggestions and comments may help the authors enhancing the draft of their valuable manuscript:

1. Title and Abstract:

Title: The title succinctly encapsulates the main topic of the review. However, consider rephrasing to enhance clarity and impact, for example, "Evaluating Thromboprophylaxis Strategies for High-Risk Pregnancy: A Current Perspective."

Abstract:

  • Completeness: The abstract provides an appropriate summary of the manuscript. Ensure that it includes objectives, methods, results, and conclusions to give readers a complete overview.
  • Clarity: Improve the clarity by simplifying complex sentences and avoiding jargon.
  •  

2. Introduction:

  • Background: Adequately provided but expand on the historical context and the evolution of current thromboprophylaxis practices.
  • Research Gap: While implied, explicitly state the gap in the existing literature your review aims to fill.
  • Objectives: Clearly define the objectives of your review. What specific aspects of thromboprophylaxis in high-risk pregnancies are you examining?

3. Literature Review:

  • Scope and Depth: While comprehensive, the review tends to list studies rather than synthesize their findings. Aim for a narrative that combines similar studies, contrasting different approaches and outcomes.
  • Critical Analysis: Increase the depth of your review by critically analyzing the methodologies and findings of the cited studies. Discuss limitations, discrepancies, and the impact of study design on outcomes.
  • Current and Relevant: Ensure all referenced studies are up-to-date and relevant. Highlight new developments and how they impact current practices.

4. Methodology:

  • Details of the Search Strategy: Specify the date range of the articles reviewed, languages considered, and any exclusion criteria.
  • Selection and Evaluation Process: Describe the process used to select studies for inclusion and assess their quality. Mention any tools or criteria used for quality assessment.
  • Data Extraction and Synthesis: Elaborate on how data were extracted from the selected studies and the approach used to synthesize the information.

5. Discussion:

  • Integration of Findings: Integrate your findings with existing literature, highlighting how your review contributes new insights or corroborates previous studies.
  • Implications for Practice: Provide specific recommendations for clinical practice based on your review. How can these findings be implemented to improve patient outcomes?
  • Limitations: Discuss the limitations of your review, such as publication bias, variability in study designs, or gaps in the current research.
  • Future Research: Suggest areas for future research to address unresolved questions or emerging trends.

6. Conclusions:

  • Summary of Key Findings: Concisely restate the main findings of your review and their significance.
  • Relevance to Clinical Practice: Emphasize the practical implications of your findings for clinicians and policymakers.
  • Call to Action: End with a clear call to action, urging the application of your findings or further investigation into specific areas.

7. The manuscript demonstrates an overall competent use of English language. However, like most academic writings, there could be room for improvement to enhance clarity, coherence, and readability. Here's an evaluation based on common linguistic aspects:

  1. Grammar and Syntax: Ensure that sentences are grammatically correct and well-structured. Long, complex sentences can be broken down into shorter ones to improve clarity and readability. Pay attention to subject-verb agreement, correct use of tenses, and proper placement of modifiers to avoid ambiguity.

Ex. Original: "The research findings was significant and suggests changes."

Revised: "The research findings were significant and suggest changes."

Issue: Subject-verb agreement; "findings" is plural, so "was" should be "were."

  1. Vocabulary and Terminology: The manuscript should consistently use appropriate and specific medical terminology. However, overly complex, or technical language can alienate readers not specialized in the field. Consider defining or simplifying extremely specialized terms without diluting the scientific accuracy.

Ex. Original: "Blood problems in pregnant women are bad."

       Revised: "Hematological disorders in pregnant women present significant health risks."

Issue: Use of vague and non-academic language ("problems", "bad"); improvement with specific medical terminology.

  1. Cohesion and Coherence: The text should flow logically from one section to the next. Use transitional phrases to link ideas and paragraphs. Ensure each paragraph has a clear main idea, supported by relevant details or evidence.

Ex. Original: "Thromboprophylaxis is important. The risk factors are many. Pregnant women are at risk."

      Revised: "Thromboprophylaxis is crucial because pregnant women face numerous risk factors."

       Issue: Disjointed sentences; improved by combining ideas for better flow.

  1. Concision and Redundancy: Avoid unnecessary repetition and redundancy. Each word should contribute to the meaning of the sentence or the development of the argument. Eliminating superfluous words can make sentences more powerful and easier to understand.
  2. Consistency: Maintain consistency in terminology, abbreviations, and formatting throughout the manuscript. This includes the use of British vs. American English, units of measurement, and the formatting of dates and numbers.

Ex. Original: "It is important to note that there are many pregnant women who experience..."

Revised: "Many pregnant women experience..."

Issue: Redundant phrase "It is important to note that"; improved by getting straight to the point.

  1. Spelling and Punctuation: Check the manuscript for spelling errors and correct punctuation. Misused punctuation can change the meaning of a sentence, and spelling errors can diminish the manuscript's credibility.

Ex. Original: "Patients history of VTE effects the choice of treatment."

Revised: "A patient's history of VTE affects the choice of treatment."

Issue: Misuse of "effects" (should be "affects"), missing apostrophe in "patients".

  1. Tone and Formality: The manuscript should maintain a formal, academic tone. Avoid colloquialisms, slang, and conversational language. Passive voice is common in scientific writing, but using active voice where appropriate can make sentences more dynamic and clear.

Ex. Original: "Doctors need to watch out for blood clots in pregnant ladies."

Revised: "Physicians must monitor pregnant patients for signs of thrombosis."

Issue: Informal tone ("watch out", "ladies"); improved by using formal language and correct medical terms.

  1. Clarity and Precision: Be as clear and precise as possible in your descriptions and arguments. Avoid vague language and generalizations. Each sentence should convey a specific idea or piece of information.

Ex. Original: "Sometimes blood clots can be a problem for women who are expecting."

Revised: "Venous thromboembolism poses a significant risk to pregnant women."

Issue: Vague language ("sometimes", "can be a problem", "expecting"); improved by being specific and using proper medical terminology.

Comments on the Quality of English Language

The manuscript demonstrates an overall competent use of English language. However, like most academic writings, there could be room for improvement to enhance clarity, coherence, and readability.

Author Response

Comments and Suggestions for Authors

Review Report on the Manuscript "Current approaches for thromboprophylaxis in pregnancy with high thromboembolism risk"

The review article aims to address the crucial issue of venous thromboembolism (VTE) during pregnancy, identifying the condition as a leading cause of maternal death in developed countries. the following suggestions and comments may help the authors enhancing the draft of their valuable manuscript:

  1. Title and Abstract:

Comment:

Title: The title succinctly encapsulates the main topic of the review. However, consider rephrasing to enhance clarity and impact, for example, "Evaluating Thromboprophylaxis Strategies for High-Risk Pregnancy: A Current Perspective."

Response:

We thank the Reviewer for this practical comment. Based on it, we changed the title to make it as exact, as possible.

Abstract:

  • Completeness: The abstract provides an appropriate summary of the manuscript. Ensure that it includes objectives, methods, results, and conclusions to give readers a complete overview.

Comment:

The abstract should be a single paragraph and should follow the style of structured abstracts, but without headings

Response:

We thank also for this comment. However, according to the Instructions for Authors, „the abstract should be a single paragraph and should follow the style of structured abstracts, but without headings“.

Anyway, we tried to modify the contents of the abstract to include the required aspects, such as objectives, methods and conclusions.

  • Clarity:

Comment:

Improve the clarity by simplifying complex sentences and avoiding jargon.

Response:

We thank the Reviewer also for this useful comment. We rewrote the abstract to make it clearer.

  1. Introduction:
  • Background:

Comment:

Adequately provided but expand on the historical context and the evolution of current thromboprophylaxis practices.

Response:

Again, we thank the Reviewer for his kind advice. We included the history of the evolution of thromboprophylaxis in the Introduction with the summary in the form of Figure 2 and with the focus on pregnancy in the text.

  • Research Gap:

Comment:

While implied, explicitly state the gap in the existing literature your review aims to fill.

Response:

At the end of the Introduction, we specified the aims of the review – we aimed to fill the gap in the existing literature regarding the optimal dose of thromboprophylaxis, duration of an increased risk of VTE and the interaction of risk factors of thromboembolic complications in pregnancy.

  • Objectives:

Comment:

Clearly define the objectives of your review. What specific aspects of thromboprophylaxis in high-risk pregnancies are you examining?

Response:

We examined several specific aspects of thromboprophylaxis in high-risk pregnancies, such as the assessment of risk stratification strategies for thromboprophylaxis in pregnancy, the role of inherited thrombophilic states in this evaluation, recommendations related to the circumstances of previous VTE in pregnant women, the form and dose of anticoagulant thromboprophylaxis, monitoring of its effectiveness and also management of cesarean section and neuraxial anesthesia in the patients with anticoagulant thromboprophylaxis.

  1. Literature Review:
  • Scope and Depth:

Comment:

While comprehensive, the review tends to list studies rather than synthesize their findings. Aim for a narrative that combines similar studies, contrasting different approaches and outcomes.

Response: We thank the Reviewer for this practical advice. We summarized the most important characteristics of discussed guidelines from Tables 1, 2, 3 and 10 in the separate part of the Discussion.

  • Critical Analysis:

Comment:

Increase the depth of your review by critically analyzing the methodologies and findings of the cited studies. Discuss limitations, discrepancies, and the impact of study design on outcomes.

Response: Again, we thank the Reviewer for such an expert comment. We tried to add the limitations, information related to the influence of study design on the interpretation of its conclusions and their strengths that are highlighted in red bold text in the paragraphs 3.1-3.20.

  • Current and Relevant:

Comment:

Ensure all referenced studies are up-to-date and relevant. Highlight new developments and how they impact current practices.

Response:

From 72 originally used sources of literature, 45 were published or cited during the last five years, so they should represent the most actual findings. Moreover, we highlighted the new developments and influence of their findings on the current recommendations e.g. in association to the diagnostics of prothrombotic risk (reference 4 in the Introduction), pathophysiology of varicose veins (section 3.8) or treatment (reference 40 in the section 3.10). References related to the risk stratification strategies for thromboprophylaxis in pregnancy published by SFOG, QCG and RCOG and summarized in Table 1 should also be the most actual.

Recommendations of ASH, ACOG and SOGC mentioned in Tables 7, 8, 9, 10 and 12 were also published during the last 5 years.

  1. Methodology:
  • Details of the Search Strategy:

Comment:

Specify the date range of the articles reviewed, languages considered, and any exclusion criteria.

Response:

As mentioned above, wherever possible, we aimed to review the information from the articles published during the last five years. However, some facts revealed in the Introduction or in the section 3 were reported earlier, but they are still valid. This is the reason why we left them in the manuscript. We took into account the articles found by the search in databases PubMed, Web of Science and Scopus, written mainly in English language, but also with the note that it was translated from another original language. We excluded all the studies  performed only in nonpregnant population.

  • Selection and Evaluation Process:

Comment:

Describe the process used to select studies for inclusion and assess their quality. Mention any tools or criteria used for quality assessment.

Response:

Thank you for this helpful comment. In addition to the previous responses related to the search strategy, regarding the quality assessment, we preferred the information obtained in randomized controlled trials which avoid confounding and minimize selection bias and systematic reviews which indicate which studies are deemed the most reliable. This information was added in the second paragraph of the section 2 Methods.

  • Data Extraction and Synthesis:

Comment:

Elaborate on how data were extracted from the selected studies and the approach used to synthesize the information.

Response:

We focused on the information valid for pregnant patients with the risk of thromboembolic complications and the need of anticoagulant thromboprophylaxis, so we selected studies performed in this population. Subsequently, we synthesized the information according to the particular risk factor in the section 3, circumstances of the use of antithrombotic prophylaxis in section 4, the presence of thrombophilia in section 5 or the studies discussing the doses of thromboprophylaxis in section 6 of the article.

  1. Discussion:
  • Integration of Findings:

Comment:

Integrate your findings with existing literature, highlighting how your review contributes new insights or corroborates previous studies.

Response:

We thank the Reviewer for another comment aiming to achieve complex insight into our reviewed topic. Based on it, we added the concluding paragraph corroborating previous studies after the narrative summary and before the section dedicated to the recommendations for the clinical practice.

  • Implications for Practice:

Comment:

Provide specific recommendations for clinical practice based on your review. How can these findings be implemented to improve patient outcomes?

Response:

The recommendations for clinical practice based on our review could be the following:

- to evaluate the need of thromboprophylaxis as soon, as the pregnancy is confirmed, consider the presence of the prothrombotic risk factors related to the comorbidities and circumstances of previous VTE episode

- in patients with previous idiopathic, recurrent or hormone-related VTE, use ante- and also postpartum prophylaxis

- in the patients with VTE associated with a major reversible risk factor / without thrombophilic state, use postpartum thromboprophylaxis

- in pregnant patients with prothrombotic risk factors undergoing cesarean section, consider postpartum thromboprophylaxis

- control the changes in the health condition of the patient and modify the actual thromboprophylaxis on the individual basis

- evaluate the presence of bleeding symptoms or allergic reaction, regularly assess platelet count, function of antithrombin, renal parameters and liver function tests

- in pregnant women with extreme body weight, renal impairment, recurrency of VTE or in the suspicion of incompliance, evaluate anti-Xa activity of LMWH

- to take the control over the prothrombotic comorbidities and complications developed during pregnancy, prefer the multidisciplinary approach

- along with the anticoagulant thromboprophylaxis, use nonpharmacologic prophylaxis, such as intermittent pneumatic compression or elastic stockings

  • Limitations:

Comment:

Discuss the limitations of your review, such as publication bias, variability in study designs, or gaps in the current research.

Response:

We thank the Reviewer for the pragmatic and useful comment. Limitations of the presented review are the inclusion of the studies with dif-ferent design – prospective and retrospective studies, randomized controlled trials and also sys-tematic reviews. Moreover, we sincerely admit that the review con-tains data representing the selection bias caused by the inclusion of the studies with the absence of the control group. However, they only confirm the conclusions of the randomized controlled trials and systematic reviews used in this review, as well.

We added this comment after the section about the recommendations for clinical practice.

  • Future Research:

Comment:

Suggest areas for future research to address unresolved questions or emerging trends.

Response:

We modified and moved the paragraph entailing the issues related to the future research from Conclusion to the last paragraph of Discussion, as proposed by the Reviewer.

  1. Conclusions:
  • Summary of Key Findings:

Comment:

Concisely restate the main findings of your review and their significance.

Response:

As suggested by the Reviewer, in the first paragraph of the Conclusion, we concisely restated the main findings of our review.

  • Relevance to Clinical Practice:

Comment:

Emphasize the practical implications of your findings for clinicians and policymakers.

Response:

We thank the Reviewer also for this note. Based on it, we added the second paragraph of the Conclusion. Due to the requirement of further Reviewers to be concise in this section, the paragraph entailing the practical implication consists of one sentence. Hopefully, it is sufficient.

  • Call to Action:

Comment:

End with a clear call to action, urging the application of your findings or further investigation into specific areas.

Response:

Due to this practical comment of the Reviewer, we modified the Conclusion.

  1. The manuscript demonstrates an overall competent use of English language. However, like most academic writings, there could be room for improvement to enhance clarity, coherence, and readability. Here's an evaluation based on common linguistic aspects:

Comments:

  1. Grammar and Syntax:

Ensure that sentences are grammatically correct and well-structured. Long, complex sentences can be broken down into shorter ones to improve clarity and readability. Pay attention to subject-verb agreement, correct use of tenses, and proper placement of modifiers to avoid ambiguity.

Ex. Original: "The research findings was significant and suggests changes."

Revised: "The research findings were significant and suggest changes."

Issue: Subject-verb agreement; "findings" is plural, so "was" should be "were."

  1. Vocabulary and Terminology: The manuscript should consistently use appropriate and specific medical terminology. However, overly complex, or technical language can alienate readers not specialized in the field. Consider defining or simplifying extremely specialized terms without diluting the scientific accuracy.

Ex. Original: "Blood problems in pregnant women are bad."

       Revised: "Hematological disorders in pregnant women present significant health risks."

Issue: Use of vague and non-academic language ("problems", "bad"); improvement with specific medical terminology.

  1. Cohesion and Coherence: The text should flow logically from one section to the next. Use transitional phrases to link ideas and paragraphs. Ensure each paragraph has a clear main idea, supported by relevant details or evidence.

Ex. Original: "Thromboprophylaxis is important. The risk factors are many. Pregnant women are at risk."

      Revised: "Thromboprophylaxis is crucial because pregnant women face numerous risk factors."

       Issue: Disjointed sentences; improved by combining ideas for better flow.

  1. Concision and Redundancy: Avoid unnecessary repetition and redundancy. Each word should contribute to the meaning of the sentence or the development of the argument. Eliminating superfluous words can make sentences more powerful and easier to understand.
  2. Consistency: Maintain consistency in terminology, abbreviations, and formatting throughout the manuscript. This includes the use of British vs. American English, units of measurement, and the formatting of dates and numbers.

Ex. Original: "It is important to note that there are many pregnant women who experience..."

Revised: "Many pregnant women experience..."

Issue: Redundant phrase "It is important to note that"; improved by getting straight to the point.

  1. Spelling and Punctuation: Check the manuscript for spelling errors and correct punctuation. Misused punctuation can change the meaning of a sentence, and spelling errors can diminish the manuscript's credibility.

Ex. Original: "Patients history of VTE effects the choice of treatment."

Revised: "A patient's history of VTE affects the choice of treatment."

Issue: Misuse of "effects" (should be "affects"), missing apostrophe in "patients".

  1. Tone and Formality: The manuscript should maintain a formal, academic tone. Avoid colloquialisms, slang, and conversational language. Passive voice is common in scientific writing, but using active voice where appropriate can make sentences more dynamic and clear.

Ex. Original: "Doctors need to watch out for blood clots in pregnant ladies."

Revised: "Physicians must monitor pregnant patients for signs of thrombosis."

Issue: Informal tone ("watch out", "ladies"); improved by using formal language and correct medical terms.

  1. Clarity and Precision: Be as clear and precise as possible in your descriptions and arguments. Avoid vague language and generalizations. Each sentence should convey a specific idea or piece of information.

Ex. Original: "Sometimes blood clots can be a problem for women who are expecting."

Revised: "Venous thromboembolism poses a significant risk to pregnant women."

Issue: Vague language ("sometimes", "can be a problem", "expecting"); improved by being specific and using proper medical terminology.

Response:

We thank the Reviewer for these helpful and illustrative comments on formal inaccuracies. We tried to apply the changes in the whole manuscript, as advised.

We thank the Reviewer for for all the complex and expert comments listed above. We sincerely appreciate each note contributing to the improvement of our manuscript and hope that the manuscript is now improved sufficiently.

Comments on the Quality of English Language

The manuscript demonstrates an overall competent use of English language. However, like most academic writings, there could be room for improvement to enhance clarity, coherence, and readability.

Reviewer 2 Report

Comments and Suggestions for Authors

This manuscript provides a comprehensive overview of the impact of pregnancy on the risk of VTE and summarizes current approaches to thromboprophylaxis. It covers various risk factors, including previous VTE, family history, smoking, obesity, and more. However, clarity could be improved in some sections. Additionally, addressing the following questions might enhance the manuscript:

  1. Can you elaborate on the physiological changes during pregnancy influencing coagulation factors?
  2. What specific recommendations are provided for managing anticoagulant thromboprophylaxis in patients undergoing cesarean section?
  3. Could the paper discuss potential limitations or gaps in existing thromboprophylaxis strategies?
  4. The use of LMWH is highlighted, but are there alternative anticoagulant options considered for specific patient groups during pregnancy?
  5. How do the findings contribute to the existing knowledge on VTE risk in pregnant and postpartum women?
  6. I suggest the authors not to use the abbreviations in the headings of the manuscript.
  7. The should be difference between the font sizes of the text and the table and figure legends.

Author Response

Comments and Suggestions for Authors

This manuscript provides a comprehensive overview of the impact of pregnancy on the risk of VTE and summarizes current approaches to thromboprophylaxis. It covers various risk factors, including previous VTE, family history, smoking, obesity, and more. However, clarity could be improved in some sections. Additionally, addressing the following questions might enhance the manuscript:

  1. Comment:

Can you elaborate on the physiological changes during pregnancy influencing coagulation factors?

Response:

The development of physiological changes during pregnancy influencing coagulation factors is described in paragraph 3. Based on this comment, we highlighted the procoagulation state developed during pregnancy. The outline of physiological changes during pregnancy is also provided in Figure 1.

  1. Comment:

What specific recommendations are provided for managing anticoagulant thromboprophylaxis in patients undergoing cesarean section?

Response:

The recommendations for pregnant women undergoing cesarean section are provided in Table 1, where this surgical procedure – according to ACCP guidelines - is considered as high risk factor. Additionally, according to QCG and SFOG guidelines, it represents significant factor in the evaluation of postpartum prophylaxis. Furthermore, the whole section 7.2 discusses the management of cesarean section in the patients with anticoagulant thromboprophylaxis in the specific clinical situations of emergent and elective cesarean section with and without risk factors.

  1. Comment:

Could the paper discuss potential limitations or gaps in existing thromboprophylaxis strategies?

Response:

We thank the Reviewer also for this practical comment. The limitations and areas for future research are provided in the last two paragraphs of the Discussion. Hopefully, it is sufficient in this extent.

  1. Comment:

The use of LMWH is highlighted, but are there alternative anticoagulant options considered for specific patient groups during pregnancy?

Response:

The manuscript discusses also the alternative anticoagulants, such as unfractionated heparin (please, see section 6.1 of the manuscript), fondaparinux (section 6.3), danaparoid (section 6.4), vitamin K antagonists (section 6.5) and DOACs (section 6.6). Based on this comment, we added more information about them in the Table 8.

  1. Comment:

How do the findings contribute to the existing knowledge on VTE risk in pregnant and postpartum women?

Response:

We thank the Reviewer for this useful comment. As stated in the first two paragraphs of the Conclusion, in the reviewed articles, there is an agreement in the fact that previous VTE episode and selected thrombophilia (e.g. homozygous form of factor V Leiden mutation and prothrombin variant G20210A) are the most significant prothrombotic risk factors. Therefore, in these circumstances, the international guidelines recommend thromboprophylaxis during the pregnancy and postpartum period.

The practical output of this review could be the information about the most appropriate dosing regimens and the summary of the risk of VTE in the course of pregnancy and puerperium in the context of discussed risk factors.

  1. Comment:

I suggest the authors not to use the abbreviations in the headings of the manuscript.

Response:

We thank the Reviewer for this formal comment. Based on it, we removed all the abbreviations from the headings and subheadings of the manuscript.

  1. Comment:

The should be difference between the font sizes of the text and the table and figure legends.

Response:

We are thankful the Reviewer also for this attentive comment – we admit that it is required to change the font size from 10 (in which the whole text is written) to 9, as recommended by the template. Therefore, we corrected this inaccuracy and we apologize for it. 

We thank the Reviewer for all useful comments listed above. We sincerely appreciate each note contributing to the improvement of our manuscript.

Reviewer 3 Report

Comments and Suggestions for Authors

Dear Authors!

My congratulations for performing such a beautiful and comprehensive review on a really important clinical problem. This review summarizes all the data one need to know about VTE prevention in pregnant. I believe the review will be useful for practitioners as a source of information about any aspect of the problem.

I have some minor remarks to be addressed.

Table 1. D-dimer is not an anticoagulation factor. It’s just a marker of fibrin degradation. D-dimer doesn’t play an independent role in the coagulation processes. The table descriptions have to be changed or D-dimer has to be excluded.

Line 88. Please, present not just a statement that VTE is the leading cause of maternal mortality. This must be supported by data for readers to understand the magnitude of the problem. What is the incidence of DVT and PE? It has to be presented clearly.

What are the exact risks related to multiparity, varicose veins, preeclampsia, weight gain, Caesarian section, postpartum infection? Please, present it in numbers. I believe if it just stated with no data confirming seems not to be enough.

Line 163. It’s better use varicose veins, not varicosis.

Lines 166-168. I would suggest either expand description of varicose veins mechanisms or skip it completely. Because, the number of molecules involved in it is much more than three that is now mentioned.

Line 177. I suggest to discuss multiple pregnancies and parity as risk factor in the same paragraph. And for multiple pregnancies exact risks have to be presented also.

Conclusions must be shortened to 2-3 sentences without references.

Comments on the Quality of English Language

None

Author Response

Začiatok formulára

Comments and Suggestions for Authors

Dear Authors!

My congratulations for performing such a beautiful and comprehensive review on a really important clinical problem. This review summarizes all the data one need to know about VTE prevention in pregnant. I believe the review will be useful for practitioners as a source of information about any aspect of the problem.

I have some minor remarks to be addressed.

We thank the Reviewer for this positive and encouraging words, as well as for all the helpful comments listed below. We sincerely appreciate each note contributing to the improvement of our manuscript.

Comment:

Table 1. D-dimer is not an anticoagulation factor. It’s just a marker of fibrin degradation. D-dimer doesn’t play an independent role in the coagulation processes. The table descriptions have to be changed or D-dimer has to be excluded.

Response:

We thank the Reviewer for this attentive comment. We corrected the heading of the Figure 1 from “Anticoagulation factors” to “Anticoagulation factors and markers of fibrinolysis”, as D-dimers are one of such markers of fibrinolysis.

Comment:

Line 88. Please, present not just a statement that VTE is the leading cause of maternal mortality. This must be supported by data for readers to understand the magnitude of the problem. What is the incidence of DVT and PE? It has to be presented clearly.

Response:

We thank the Reviewer for this advice and added the specification of the incidence of DVT and PE in the manuscript.

Comment:

What are the exact risks related to multiparity, varicose veins, preeclampsia, weight gain, Caesarian section, postpartum infection? Please, present it in numbers. I believe if it just stated with no data confirming seems not to be enough.

Response:

We thank the Reviewer also for this requirement for the specification. Based on it, we added mentioned information on the particular places in the text. 

Comment:

Line 163. It’s better use varicose veins, not varicosis.

Response:

We thank the Reviewer also for this note. We corrected the text, as advised.

Comment:

Lines 166-168. I would suggest either expand description of varicose veins mechanisms or skip it completely. Because, the number of molecules involved in it is much more than three that is now mentioned.

Response:

We thank the Reviewer for this expert comment. We added the list of further molecules involved in the pathogenesis of varicose veins in the paragraph 1 of the section 3.8.

Comment:

Line 177. I suggest to discuss multiple pregnancies and parity as risk factor in the same paragraph. And for multiple pregnancies exact risks have to be presented also.

Response:

We put the paragraphs about multiple pregnancies and parity together, as proposed by the Reviewer and added the exact risk of VTE in the case of multiple pregnancies.

Comment:

Conclusions must be shortened to 2-3 sentences without references.

Response:

We thank the Reviewer for this practical comment. As it can be seen in this section, we made significant changes in the structure of the Conclusion with the reduction of its extent. However, we had to add further sentences due to the recommendations of further Reviewers. Thus, hopefully, it is acceptable this way.

Comments on the Quality of English Language

None

Reviewer 4 Report

Comments and Suggestions for Authors

Dear Authors,

your manuscript addresses a clinical field of major importance and of major clinical relevance.

However, the main problem actually is the formal presentation of the text, which must be rearranged in depth to improve readability. This, however, also is an issue of the publisher.

a) For example table 3 simply is not readable in its present form. However, this also refers to the other tables

b) include more figures or flow charts like Fig. 1 to help the readers and increase attraction of the manuscript.

c) all abbreviations must be explained within a list at the beginning or at the start of the paper

d) the text must be adapted in that to be also understandable for clinicians without special expertise in this field.

I am convinced that the paper significantly gains in global interest if you follow these recommendations

Comments on the Quality of English Language

I recommend to include a nativ speaker or a specialist in English language for optimization of the language. However, the main problem actually is the outlay.

Author Response

Dear Authors,

your manuscript addresses a clinical field of major importance and of major clinical relevance.

However, the main problem actually is the formal presentation of the text, which must be rearranged in depth to improve readability. This, however, also is an issue of the publisher.

Comment:

  1. a) For example table 3 simply is not readable in its present form. However, this also refers to the other tables

Response:

We thank the Reviewer for this comment. We admit that the tables are long and might be difficult to read. However, we had to comply with the template regarding the margins and style of the tables. As the manuscript should fullfil the criteria of the review article, we also tried to synthesize all the relevant information from discussed guidelines to the compex tables. Anyway, to improve the readibility of the Table 3, we deleted the information about PS deficiency, because it is the same as for PC deficiency and put them together. We would like to deeply apologize for the form of the manuscript - please, do not consider the extent of the tables and mentioned outlay as inappropriate and as the cause for the rejection of the manuscript. We made significant effort to write such a comprehensive and synthesized form of all available information about this actual and important topic. Thus, I suppose that the extent could be useful for readers.

Comment:

  1. b) include more figures or flow charts like Fig. 1 to help the readers and increase attraction of the manuscript.

Response:

We thank the Reviewer for another practical comment. Based on it, we included two more figures in the manuscript.

Comment:

  1. c) all abbreviations must be explained within a list at the beginning or at the start of the paper

Response:

We thank the Reviewer for this comment, as well. Except the explanation of the particular abbreviations in the situation of their first use in the text and in the Figures, we added the list of abbreviations at the beginning of the manuscript.

Comment:

  1. d) the text must be adapted in that to be also understandable for clinicians without special expertise in this field.

Response:

We tried to improve the text as much, as possible with many significant changes, the addition of strengths and limitations of the studies cited in the manuscript, advances in the pathophysiology of VTE, narrative summary of the guidelines, recommendations for clinical practice, limitations and also areas of future research.

I am convinced that the paper significantly gains in global interest if you follow these recommendations

Comments on the Quality of English Language

I recommend to include a nativ speaker or a specialist in English language for optimization of the language. However, the main problem actually is the outlay.

We thank the Reviewer for for all the complex and expert comments listed below. We sincerely appreciate each note contributing to the improvement of our manuscript.

Round 2

Reviewer 1 Report

Comments and Suggestions for Authors

The authors addressed all comments and enhanced the readability of the manuscript. They made great efforts. 

Comments on the Quality of English Language

Minor editions needed

Reviewer 4 Report

Comments and Suggestions for Authors

Dear authors

the revised manuscript has been improved. As a narrative review of a clinically very important issue I think the manuscript is ready for publication

Bernhard Rauch

Comments on the Quality of English Language

The English language should be checked by a professional translater or a native speaker before publication